# PID-controlled Langevin Dynamics for Faster Sampling of Generative Models

**Hongyi Chen**[1,3]*, **Jianhai Shu**[2]*, **Jingtao Ding**[2]†, **Yong Li**[2], **Xiao-Ping Zhang**[1]†

[1] Shenzhen Key Laboratory of Ubiquitous Data Enabling Laboratory,
Shenzhen International Graduate School, Tsinghua University,
Shenzhen 518055, China
[2] Department of Electronic Engineering, Tsinghua University,
Beijing 100084, China
[3] Pengcheng Laboratory, Shenzhen 518055, China

## Abstract

Langevin dynamics sampling suffers from extremely low generation speed, fundamentally limited by numerous fine-grained iterations to converge to the target distribution. We introduce PID-controlled Langevin Dynamics (PIDLD), a novel sampling acceleration algorithm that reinterprets the sampling process using control-theoretic principles. By treating energy gradients as feedback signals, PIDLD combines historical gradients (the integral term) and gradient trends (the derivative term) to efficiently traverse energy landscapes and adaptively stabilize, thereby significantly reducing the number of iterations required to produce high-quality samples. Our approach requires no additional training, datasets, or prior information, making it immediately integrable with any Langevin-based method. Extensive experiments across image generation and reasoning tasks demonstrate that PIDLD achieves higher quality with fewer steps, making Langevin-based generative models more practical for efficiency-critical applications. The implementation can be found at https://github.com/tsinghua-fib-lab/PIDLD.

## 1  Introduction

Langevin dynamics has emerged as a fundamental sampling technique for generating samples across various implicit generative models, including Energy-based Models (EBMs) [8], Score-based generative models (SGMs) [33, 34, 35], and Diffusion models [15]. This sampling method gradually approaches the target distribution through stochastic walks along energy gradients (or score functions), visualized as particles moving under the influence of energy potential field gradients, while being affected by random fluctuations that correlate with step size.

Despite its theoretical elegance and broad applicability, Langevin sampling suffers from a critical limitation: extremely low generation speed that necessitates multiple fine-grained iterations of the discretized Langevin dynamics process (e.g., NCSNv2 requires 1000+ Neural Function Evaluations (NFE) for image sampling [34]). This stands in contrast to GANs [17, 18], which enable efficient one-step sampling. During sampling, particles often encounter regions with near-zero gradient "forces" (such as local minima or unstable equilibria), necessitating the addition of random noise to escape these regions, which consequently increases the required number of sampling steps (Fig.1, left). While reducing sampling steps and increasing step sizes seems like a straightforward acceleration approach, it inevitably introduces larger noise fluctuations that significantly degrade sampling quality [23, 24].

---

*Equal Contribution.
†Corresponding Author.

39th Conference on Neural Information Processing Systems (NeurIPS 2025).

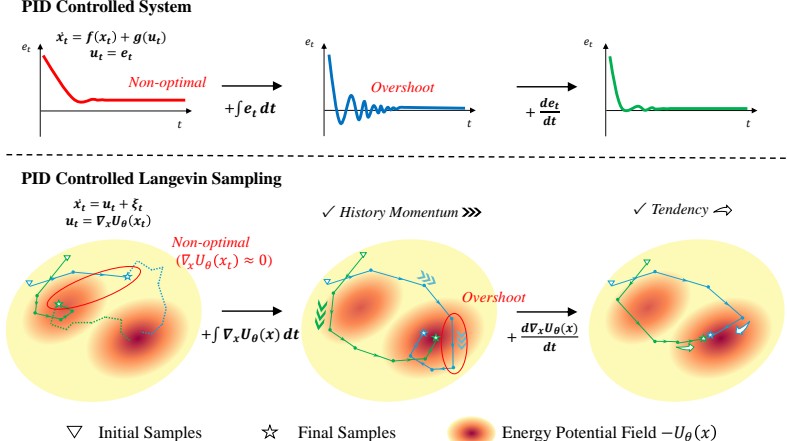

Figure 1: PID-controlled systems eliminate errors ($e_t$) at steady-state through the integral term while dampening overshoot via the derivative term. In the Langevin sampling context, we reinterpret PID's value: vanilla Langevin samples can be trapped at non-optimal equilibrium points with zero gradients, requiring noise $\xi_t$ and additional iterations to escape (dashed lines). The integral term provides a momentum-like effect that helps samples quickly escape these suboptimal points, though it introduces overshoot. The derivative term adds awareness of real-time tendencies, accelerating descent while reducing overshoot. Together, these mechanisms enhance both sampling speed and quality.

Inspired by control theory, we reinterpret the Langevin sampling process by conceptualizing energy gradients as feedback signals, establishing a correspondence between sampling and feedback control systems [10]. This approach treats particles as dynamic systems under designed feedback control rather than passive stochastic entities, allowing us to efficiently guide sampling toward high-probability regions while substantially reducing required iterations.

Specifically, we innovatively consider the historical dynamics and tendency information of potential-field gradient signals (feedback signals) during sampling. We thus propose integrating Langevin sampling with the prominent PID (Proportional-Integral-Derivative) [1] technique from feedback control to enhance and accelerate sampling in the following aspects. **Firstly,** gradient history (the integral term) creates momentum-like effects by accumulating gradient information, providing sustained directional forces that reduce ineffective random walks. These forces form inertial properties that help sample particles traverse energy barriers and unstable equilibria in complex multimodal distributions, thereby exploring more critical regions and significantly accelerating convergence to minimum energy states of the distribution [23] (Fig.1, middle). **Secondly,** gradient tendency (the derivative term) dynamically adjust sampling by responding to real-time gradient change rates, accelerating movement toward directions of consistently decreasing energy gradients while suppressing movement where gradients consistently increase, effectively mitigating overshoot issues by anticipating and counteracting excessive momentum, enabling the overall sampling process to reach potential wells rapidly and stably, thereby improving sampling speed (Fig.1, right).

In this work, we make the following contributions:

(1) By combining PID control methodology with Langevin dynamics, we propose a novel Langevin sampling acceleration algorithm: PIDLD (**P**roportional-**I**ntegral-**D**erivative **L**angevin **D**ynamics). It offers a brand new insight into applying control theory in the sampling domain. Our method is simple, easy to implement, requires no additional datasets or prior information, and eliminates the need for retraining, making it compatible with any method that uses Langevin dynamics sampling.

(2) Through theoretical analysis and empirical toy experiments, we establish the convergence and effectiveness of our algorithm, validating the reliability of the proposed sampling method.

(3) We further conduct multiple comparative experiments on Langevin sampling-based models (EBMs, SGMs) across various tasks (images, solutions of reasoning tasks), demonstrating that our proposed sampler generates faithful samples faster. In particular, PIDLD achieves at least 10× sampling speedup over baselines under SGM model.

## 2 Background and Related Work

### 2.1 Langevin Dynamics in Generative Models

Langevin dynamics (LD) is a mathematical framework describing particle motion in a potential energy field with thermal fluctuations, and has been widely applied in molecular dynamics simulations, statistical physics, and Bayesian inference [36, 26, 39]. Due to its convergence properties, it plays a crucial role in sampling for both Energy-based Models (EBMs) and Score-based Generative Models (SGMs).

EBMs have evolved significantly over the past decade, with seminal contributions from Restricted Boltzmann Machines [14], discriminative EBMs [20], and development of improved training techniques for deep EBMs [8]. These models represent probability distributions in the form $p_\theta(x) = e^{-f_\theta(x)}/Z_\theta$, where $f_\theta(x)$ is the energy function parameterized by $\theta$, and $Z_\theta$ is the normalizing constant. For sampling from EBMs, perhaps the most popular sampling algorithm is Unadjusted Langevin dynamics (ULA) [29, 8, 27]: $x_{t+1} = x_t - \epsilon \nabla_x f_\theta(x_t) + \sqrt{2\epsilon}\xi_t$, where $x_t$ is the state at iteration $t$, $\epsilon$ is the step size, and $\xi_t$ is standard Gaussian noise. Starting from a randomly chosen $x_0$, we can generate a sample by iterating the above Markov transition for a specified number of steps $T$.

SGMs originated from Song and Ermon's Noise-Conditional Score Networks (NCSN) [33], which established the core paradigm of score matching at multiple noise scales $\{\sigma_i\}_{i=1}^n$. Their subsequent NCSNv2 [34] improved stability through noise scaling and architectural improvements, while NCSN++ [35] unified score-based and diffusion approaches via stochastic differential equations. Unlike energy models, SGMs learn the gradient of the log probability density function of the noise corrupted distribution $p(\hat{\mathbf{x}}|\mathbf{x}) := \mathcal{N}(\hat{\mathbf{x}}; \mathbf{x}, \sigma_i \mathbf{I})$ through a family of score-matching methods [16] ($\mathbf{x}$ follows the underlying true data distribution), yielding the score function: $s_\theta(x, \sigma_i) = \nabla_x \log p_\theta(x, \sigma_i)$. After training the score function, we can sample from the model using annealed Langevin dynamics (ALD) [33]: $x_{t+1}^i = x_t^i + \epsilon_i \nabla_x \log p_\theta(x_t, \sigma_i) + \sqrt{2\epsilon_i}\xi_t$, where $\epsilon_i$ is relevant to the noise schedule. Notably, under the energy model assumption, $\nabla_x \log p_\theta(x) = -\nabla_x f_\theta(x) - \nabla_x Z_\theta = -\nabla_x f_\theta(x)$. Therefore, the sampling processes are essentially identical, and we will use $U_\theta(x)$ to represent both $\log p_\theta(x)$ and $-f_\theta(x)$ in subsequent discussions. The Langevin dynamics in both methods work in that as $T \to +\infty$, the distribution of $x_t$ converges to a steady-state distribution [11].

Recent research has focused on accelerating Langevin sampling. Dockhorn et al. [3] enhanced SGMs by incorporating Hamiltonian Monte Carlo methods [26] and proposed critically-damped Langevin diffusion (CLD) based SGMs that demonstrated remarkable performance improvements. However, this approach necessitates additional learning of the diffusion velocity. In contrast, our method is learning-free. Ma et al. [23, 24] introduced a matrix preconditioning technique that equalizes the curvature rates across all directions, thereby achieving faster sampling for high-resolution images. However, this technique relies heavily on target dataset statistics for preconditioning, which restricts its generalization to data-scarce or structurally diverse domains. Our approach eliminates the need for additional prior knowledge or dataset statistics, requiring only a pre-trained energy model or score model. Wen et al. [40] and Shetty et al. [31] developed momentum-enhanced Langevin dynamics to expedite sampling. Our method treats Langevin sampling from a control perspective, delivering more stable results while enhancing interpretability. We provide a detailed comparison of related works in Appendix A.1.

### 2.2 PID controller

Feedback control represents the cornerstone of modern control theory, with the fundamental objective of stabilizing systems and enhancing their performance through continuous error correction [1]. Among the various feedback control strategies, Proportional-Integral-Derivative (PID) controllers remain one of the most widely adopted mechanisms in industrial applications due to their remarkable balance of simplicity and effectiveness. The standard PID control law is formulated as: $u(t) = K_p e(t) + K_i \int_0^t e(\tau)\mathrm{d}\tau + K_d \frac{\mathrm{d}e(t)}{\mathrm{d}t}$, where $e(t) = r(t) - y(t)$ represents the error between the desired reference signal $r(t)$ and the measured output $y(t)$. The control input $u(t)$ combines three terms with their respective tuning parameters: $K_p$ (proportional gain), $K_i$ (integral gain), and $K_d$ (derivative gain). The elegance of PID design lies in its intuitive interpretation: the proportional term responds to present errors, the integral term addresses accumulated past errors, and the derivative term anticipates future error tendency [9].

The PID framework has transcended traditional control applications to inspire innovations in deep learning architectures [25, 41] and optimizer [37, 2]. In particular, for the optimizer, Chen et al. [2] accelerate gradient descent optimization in deep learning via a PID controller, thereby achieving faster convergence and improved robustness against local optima. Our work is an innovative application of PID feedback control in generative models sampling.

# 3 Methods

To accelerate Langevin sampling, we reinterpret the method through the theoretical lens of feedback control. By leveraging fundamental principles from the field of feedback control, we then propose an innovative approach that integrates the Proportional-Integral-Derivative (PID) controller with Langevin sampling. Finally, building upon this framework, we validate the effectiveness of our method through toy experiments, analyze its performance characteristics, and formulate a comprehensive algorithmic framework.

## 3.1 Controlling Langevin Sampling

While LD serves as a crucial sampling method in generative modeling, it faces significant practical challenges. Firstly, standard LD suffers from slow convergence due to its random-walk behavior, requiring numerous iterations to reach the target distribution. Secondly, in multimodal distributions, sampling points rely heavily on stochastic noise to either overcome energy barriers between local minima or escape positions at unstable equilibrium points—critical locations with near-zero gradients between potential wells. Finally, its lack of dynamic response to gradient changes leads to unstable sampling across varying potential landscapes. Fundamentally, these limitations stem from the equilibrium-based nature of standard Langevin dynamics, which is ill-suited for the evolving dynamics of complex systems [22].

From a physical perspective, Langevin dynamics can be interpreted as a particle moving through a potential field $-U_\theta(x)$ while subject to thermal fluctuations. The standard discrete LD update corresponds to the continuous-time stochastic differential equation: $\frac{dx}{dt} = \nabla_x U_\theta(x) + \sqrt{2}\xi(t)$. In this physical system, the gradient $\nabla_x U_\theta(x)$ can be seen as the basic feedback control signal driving the particle toward high-probability regions. However, this simple feedback mechanism proves insufficient for complex high-dimensional landscapes. Drawing inspiration from control theory[2], we can enhance this physical system with a more sophisticated control force that utilizes not only the current gradient but also its gradient history (accumulated record of past gradient values, representing the integral of gradient trajectory over time) and gradient tendency (temporal derivative of the gradient, capturing the direction and rate of gradient change):

$$\frac{dx}{dt} = u_{\text{control}}(\nabla_x U_\theta(x), \text{Gradient History}, \text{Gradient Tendency}) + \sqrt{2}\xi(t) \tag{1}$$

This enhanced control approach directly addresses the key challenges of Langevin dynamics through the following mechanisms:

- **For sampling efficiency:** Accumulated gradient history creates a momentum-like effect, providing a persistent directional force that reduces random walk behavior and accelerates convergence. Meanwhile, gradient tendency information allows the system to adapt its movement strength based on local landscape properties, accelerating movement towards directions where gradients consistently increase, thereby further improving efficiency.

- **For energy barrier and unstable equilibria:** The accumulated gradient history forms an inertial force that helps particles traverse energy barriers and unstable equilibria between modes. This substantially reduces reliance on random noise for mode exploration and enables more systematic discovery of all significant modes in a multimodal distribution[2].

- **For dynamic response:** Gradient tendency information captures the rate of change in the potential landscape, allowing the system to stabilize its behavior in different regions dynamically. When the gradient changes abruptly, this term provides damping effects that reduce unnecessary overshoot [1].

The Proportional-Integral-Derivative (PID) controller offers an ideal framework for implementing this enhanced control approach. With the gradient $\nabla_x U_\theta(x)$ as the error signal, each component

serves a distinct purpose: the proportional term provides basic gradient guidance, the integral term accumulates gradient history to create persistent acceleration and barrier-penetration capabilities, and the derivative term captures gradient change rates to enable adaptive response to different landscape regions [1]. Based on this framework, we propose integrating a PID controller into Langevin dynamics:

$$x_{t+1} = x_t + \epsilon \left( k_p \nabla_x U_\theta(x_t) + \frac{k_i}{t} \sum_{s=0}^{t} \nabla_x U_\theta(x_s) + k_d(\nabla_x U_\theta(x_t) - \nabla_x U_\theta(x_{t-1})) \right) + \sqrt{2\epsilon}\xi_t,$$

(2)

where $k_p$, $k_i$, and $k_d$ are the proportional, integral, and derivative coefficients, respectively. Empirically, to prevent the integral term from dominating the sampling process over time, we apply a $\frac{1}{t}$ normalization factor. This balances the magnitudes of P, I, and D terms, maintaining system stability while ensuring similar influence across all components by default.

To demonstrate the effectiveness of our proposed PID-controlled Langevin dynamics, we conducted toy experiments (details in Appendix C.1) on a two-dimensional Gaussian mixture distribution. Specifically, we used NCSN [33] to learn the distribution $p_{\text{data}} = \frac{1}{5}\mathcal{N}((-5,-5), I) + \frac{4}{5}\mathcal{N}((5,5), I)$ and compare our sampling method against the vanilla ALD. As Fig 2 shows, our method generates samples that more accurately match the actual distribution, evidenced by lower KL divergence. It also confirms that both integral and derivative terms accelerate convergence and enable the sampling to achieve a lower KL divergence after convergence.

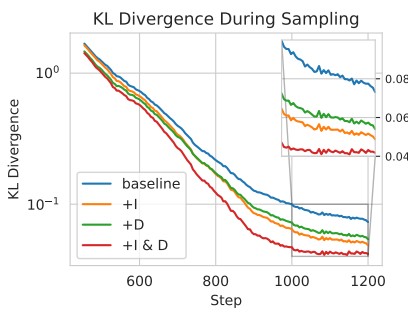

Figure 2: KL divergence along sample steps using vanilla LD(baseline), LD with integral term(+I), LD with derivative term(+D) and the complete PID controlled LD(+I&D). We use $k_p = 1, k_i = 0.1, k_d = 6$.

## 3.2 Method Analysis

**Analysis of parameter effect.** To quantitatively assess the effects of the newly added terms (integral, derivative) on sampling efficacy, we extended our analysis using the toy experiment from Section 3.1. We set $k_p = 1$ to correspond with standard Langevin dynamics, while examining how varying values $k_i$ and $k_d$ influence distributional convergence measured by the KL divergence.

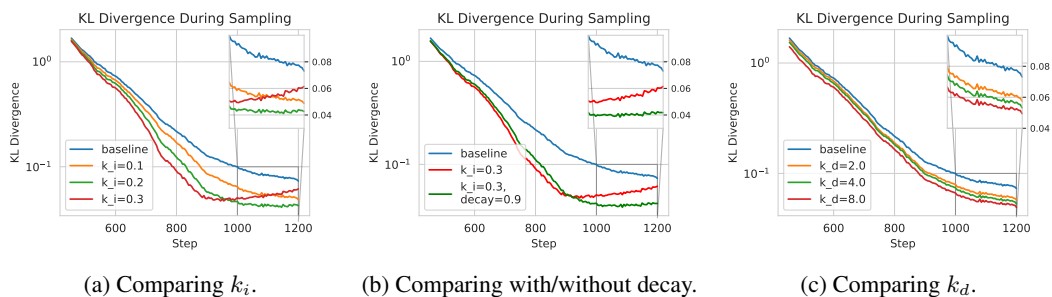

(a) Comparing $k_i$.   (b) Comparing with/without decay.   (c) Comparing $k_d$.

Figure 3: KL divergence during sampling with different parameter configurations.

As illustrated in Fig. 3a, within a specific range ($k_i \in \{0, 0.1, 0.2\}$), introducing progressively increasing integral control terms facilitates faster convergence of the KL divergence while reaching lower terminal values, suggesting that the integral term effectively assists sampling points in traversing energy barriers and unstable equilibria, thereby enhancing sampling quality. However, when $k_i$ reaches 0.3, we observe a rebound phenomenon in the KL divergence, indicating failure to converge to the stationary distribution.

To address this limitation, we implement a time-dependent decay coefficient for the integral term: $k_i(t) = \gamma^t \cdot k_i$, where $t$ denotes the sampling step and $\gamma < 1$ represents the decay rate. This mechanism stands as a principled implementation of continuous gain scheduling [19], a classic

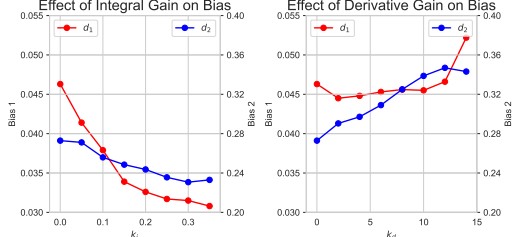
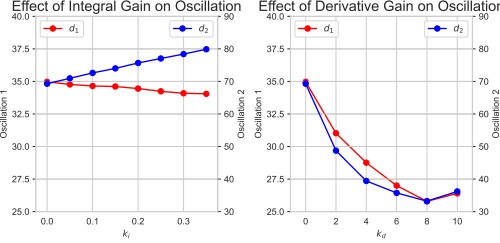

(a) Effect of $k_i$ and $k_d$ on bias. Adding $I$ term (left) excels in mitigating bias.

(b) Effect of $k_i$ and $k_d$ on oscillation. Adding $D$ term (right) excels in reducing oscillation.

Figure 4: Effect validation on $I$ term and $D$ term. The result for each hyperparameter setting is averaged over 100 independent runs with different random seeds.

control strategy well-suited for the time-varying nature of Langevin sampling. The process requires conflicting gains: a high integral gain is beneficial in the early exploration phase to build momentum and traverse energy barriers, but becomes detrimental in the late convergence phase, causing the instability seen in Fig. 3a.

Our exponential decay acts as an effective gain schedule, smoothly transitioning the controller's behavior from aggressive exploration to stable convergence by ensuring the integral term's influence diminishes as equilibrium is approached. Critically, this design guarantees that the sampling dynamics asymptotically reduce to standard Langevin dynamics ($k_i(t) \rightarrow 0$), thus ensuring theoretical convergence. As confirmed by results for $k_i = 0.3$ and $\gamma = 0.9$ in Fig. 3b, this principled approach preserves the initial acceleration while resolving the instability.

Further validation on Computer Vision tasks is provided in Appendix D.3.

From Fig. 3c, we observe that introducing a larger D term facilitates faster convergence of the KL divergence to lower values without exhibiting the rebound phenomenon. This indicates that the D term effectively helps sampling points navigate around distribution barriers, enabling them to converge more rapidly to lower points in the energy landscape representing the distribution within a finite number of steps, thereby enhancing sampling efficiency.

**Analysis of control-theoretic effect.** To empirically validate the control-theoretic effect [1] of the integral term on mitigating bias and the derivative term on reducing oscillation, we conducted further experiments (description of settings and metrics can be found in Appendix D.1 and Appendix D.2). For the bias-mitigating effect, we manually add a constant perturbation to the gradient and run annealed Langevin dynamics sampling. From Fig 4a, we observe that increasing $k_i$ helps to decrease $d_1$ and $d_2$, a measure of the final bias of cluster centers, while increasing $k_d$ yields the opposite effect, which demonstrates it is the integral term that makes the sampling algorithm robust to gradient bias. For oscillation-reducing effect, we run the toy experiments with different $k_i$ and $k_d$ again, and record the summed Euclidean distance to the final cluster centers, $d_{\text{sum}}^{(1)}$ and $d_{\text{sum}}^{(2)}$, as a measure of oscillation. From Fig 4b, we observe that $d_{\text{sum}}^{(1)}$ and $d_{\text{sum}}^{(2)}$ decrease as $k_d$ increases, while they do not show a comparable decreasing trend as we increase $k_i$. The result echoes the PID control theoretic insight that the derivative term has the advantage of reducing the oscillation in the sampling process.

**Analysis of convergence guarantee.** To provide theoretical justification for the stabilizing effect of the derivative term, we analyze its impact on convergence under the standard assumption of local strong convexity—a tractable setting for studying behavior near a potential minimum. We formally establish that the D-term preserves the convergence guarantees of standard Langevin dynamics, confirming it acts as a proper stabilizing component without disrupting the sampling process:

**Proposition 1** (Stability of Langevin Dynamics with Derivative Term)**.** *Consider the discrete-time modified Langevin dynamics:*

$$x_{t+1} = x_t + \epsilon \left( \nabla_x U_\theta(x_t) + k_d(\nabla_x U_\theta(x_t) - \nabla_x U_\theta(x_{t-1})) \right) + \sqrt{2\epsilon}\xi_t, \qquad (3)$$

*where $-U_\theta(x)$ is $m$-strongly convex ($\nabla^2 U_\theta(x) \leq -mI$), $\xi_t \sim \mathcal{N}(0, I)$, and $\epsilon, k_d > 0$. In the deterministic case ($\xi_t \equiv 0$), the system is asymptotically stable at $x^*$ (where $\nabla U_\theta(x^*) = 0$) if*

$$\epsilon < \frac{1}{(1 + 2k_d)m}. \qquad (4)$$

---

**Algorithm 1** PIDLD

---

**Require:** Score function $\nabla_x U_\theta(x) = \nabla_x \log p_\theta(x) = \nabla_x(-f_\theta(x))$, number of steps $T$, step size $\epsilon$, control parameters $k_p, k_i, k_d$, decay rate $\gamma < 1$, initial point $x_0$
1: Initialize integral term $I_0 = \mathbf{0}$
2: Compute initial score $s_0 = \nabla_x U_\theta(x_0)$
3: **for** $t = 0$ to $T - 1$ **do**
4:      $s_t = \nabla_x U_\theta(x_t)$
5:      $P_t = s_t$ {Proportional term}
6:      $I_t = \frac{1}{t+1} \cdot (I_{t-1} \cdot t + s_t)$ {Integral term}
7:      $D_t = (s_t - s_{t-1})$ {Derivative term}
8:      $u_t = k_p P_t + k_i I_t + k_d D_t$ {Compute control signal}
9:      $x_{t+1} = x_t + \epsilon \cdot u_t + \sqrt{2\epsilon} \cdot \xi_t$, where $\xi_t \sim \mathcal{N}(0, I)$ {Update state}
10:      $k_i = k_i \cdot \gamma$
11: **end for**
12: **return** $\hat{x} = x_T$

---

*Then with the random noise ($\xi_t \neq 0$), the system admits a unique stationary distribution, i.e. $x_\infty \sim \pi^*$.*

The proof is provided in Appendix B. This proposition establishes that under appropriate step size and $k_d$ constraints, our modified Langevin dynamics preserves the fundamental convergence property. Specifically, in locally strongly convex energy landscapes, convergence to the stationary distribution is ensured with properly calibrated parameters.

### 3.3 The Proposed Algorithm

Based on our empirical findings and theoretical analyses, we propose **PIDLD** (**P**roportional-**I**ntegral-**D**erivative **L**angevin **D**ynamics, Algorithm 1), a novel sampling algorithm that effectively combines the strengths of PID control theory with Langevin dynamics. This approach aims to address the limitations of traditional Langevin-based samplers, particularly in efficiently navigating complex energy landscapes characterized by multiple modes and energy barriers.

An application of PIDLD's training-free, modular design is its seamless integration with annealed Langevin Dynamics (ALD) [33]. This integration preserves ALD's beneficial annealing structure while enhancing efficiency at each noise level. The implementation is straightforward: the PIDLD controller replaces the vanilla Langevin steps within each noise level $\sigma_i$. Crucially, to maintain a smooth trajectory across noise scales, the final accumulated $I_t$ and gradient ($\nabla_x U_\theta(x_t)$ for the derivative term) from the previous scale are carried over as the initial state for the next scale $\sigma_{i-1}$. This ensures that historical information is leveraged continuously throughout the sampling process.

## 4 Experiments

In the experiments, we evaluated our method against standard Langevin sampling on mainstream generative models (SGM, EBM). We focused on the effectiveness in classical image generation tasks and investigated PIDLD's efficacy in reasoning task solutions via energy model sampling.

### 4.1 Image Sampling for Generation Tasks

In this section, we evaluate how off-the-shelf generative models that use an LD-based sampling method can be accelerated by the proposed PIDLD whilst maintaining, or even enhancing, image sampling quality, without retraining or post-training the base model.

**Datasets and Base Models.** Following NCSN [33], we test image generation quality on CIFAR10 (32×32) and CelebA (64×64) datasets (both are unconditional datasets). For the score-based generative model (SGM), we adopt the NCSNv2 [34] architecture and initialize the score network with its official pretrained checkpoint. For the energy-based model (EBM), we leverage the Implicit Generative Energy-Based Model (IGEBM) [8], which has established itself as a seminal framework in the energy-based modeling literature. We initialize the model with its official pretrained checkpoint on CIFAR10 and retrain the model on CelebA. Please refer to the Appendix C.2 for detailed implementation.

Table 1: **FID** comparison for SGM and EBM models across different datasets and NFEs. Note that NCSNv2 uses $L$ noise levels. and in each noise level, we perform $T$ Langevin update steps, so the NFE is given by $L \times T$. The configuration where $L > T$ follows the setting in [34].

| Dataset | Method | SGM [34] | | | | | EBM [8] | | | |
|---------|--------|------|------|-------|-------|-------|------|------|------|------|
| | NFEs | 25×1 | 100×1 | 232×1 | 232×3 | 232×5 | 10 | 20 | 30 | 40 |
| CIFAR10 | Vanilla [33, 8] | 46.8 | 17.2 | 16.0 | 12.8 | 12.5 | 135.8 | 58.1 | 40.3 | 35.3 |
| | MILD [31] | / | / | 15.5 | 13.6 | 13.0 | 111.4 | 49.9 | 38.9 | 34.4 |
| | Ours | **18.3** | **12.1** | **11.7** | **11.6** | **11.4** | **99.0** | **46.1** | **32.8** | **33.2** |
| | NFEs | 50×1 | 250×1 | 500×1 | 500×3 | 500×5 | 15 | 20 | 25 | 30 |
| CelebA | Vanilla | 25.0 | 13.6 | 14.0 | 11.3 | 9.5 | 109.1 | 63.5 | 41.3 | 35.4 |
| | MILD | / | / | 9.0 | 9.4 | 11.0 | 60.1 | 41.1 | 35.3 | 32.9 |
| | Ours | **8.0** | **5.7** | **5.9** | **5.9** | **5.6** | **58.0** | **38.9** | **32.2** | **30.0** |

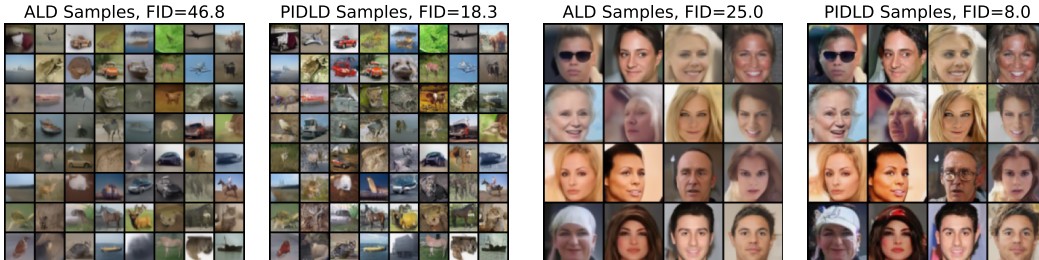

ALD Samples, FID=46.8    PIDLD Samples, FID=18.3    ALD Samples, FID=25.0    PIDLD Samples, FID=8.0

(a) CIFAR10 samples with 25 sampling steps.    (b) CelebA samples with 50 sampling steps

Figure 5: Image generation quality comparison of ALD and PIDLD under low NFEs.

**Baselines and Metrics.** We choose the vanilla ALD [33, 8] and momentum-imbued Langevin dynamics (MILD,[31]) as our baselines. Note that vanilla ALD is the degenerate version of our PIDLD algorithm when we set $k_p = 1, k_i = 0, k_d = 0$. Accordingly, we fix $k_i$ and $k_d$ to 0 and tune only $k_p$ to get the best FID score of vanilla ALD on each NFE. For MILD, we directly use the published results of NCSNv2 from the original paper, and additionally evaluate its performance in IGEBM. We report FID scores [13] computed on 10000 samples for each test.

**Experiment Results.** The results are shown in Table 1, demonstrating that PIDLD consistently outperforms both vanilla and momentum ALD across all NFEs on CIFAR10 and CelebA datasets, regardless of the base model used. Compared to the best-performing baseline, our method achieves significant performance gains of 7.6% (SGM) and 3.4% (EBM) on CIFAR10, and even more substantial improvements of 38.3% (SGM) and 8.8% (EBM) on CelebA. Notably, when using SGM, PIDLD achieves FID scores of 12.1/8.0 with only 100/50 NFEs on CIFAR10/CelebA, exceeding the best performance of baselines. This demonstrates at least a 10× speedup[3] compared to the baselines, which reach optimal performance at much larger NFEs, highlighting PIDLD's computational efficiency alongside quality improvements.

Further, we have the following findings. First, SGM outperforms EBM in absolute performance, indicating more accurate score prediction. This suggests that PIDLD's effectiveness in enhancing sampling performance and efficiency correlates directly with the accuracy of the underlying pre-trained score prediction model. Second, when compared to the baseline MILD algorithm, our approach exhibits consistently superior and more stable performance across varying inference budgets. This enhancement results from our extension of MILD's momentum-based acceleration idea through the integration of PID-based feedback control, which simultaneously incorporates both historical feedback and current trend signals. This comprehensive control mechanism substantially improves upon the robustness of purely momentum-driven methods, yielding advancements in both sampling quality and computational efficiency.

---

[3]Regarding the reasonability of using NFE as a measure of computation cost, please refer to Appendix E.

Table 2: **Accuracy** comparison for baseline and our models across harder Sudoku and Connectivity tasks [7] under different NFEs.

| Task | Method | EBM[7] Accuracy (%) | | | | | |
|------|--------|------|------|------|------|------|------|
| Sudoku | NFEs | 5 | 10 | 15 | 30 | 40 | 80 |
| | Vanilla | 45.99 | 51.00 | 50.93 | 50.77 | 53.63 | 55.02 |
| | MILD | 49.75 | 54.82 | 53.55 | 55.25 | 56.56 | 56.64 |
| | Ours | **50.54** | **55.48** | **55.55** | **55.94** | **57.02** | **56.64** |
| Connectivity | NFEs | 1 | 2 | 3 | 4 | 5 | 10 |
| | Vanilla | 86.16 | 87.22 | 87.22 | 87.48 | 87.38 | 87.49 |
| | MILD | 86.16 | 88.54 | 89.21 | 89.75 | 90.15 | 90.33 |
| | Ours | 86.16 | **91.32** | **92.31** | **92.82** | **92.95** | **93.28** |

**Visualization.** From the experiment results, we find that compared to vanilla ALD, PIDLD delivers markedly higher image quality, especially under low NFEs. We illustrate this contrast in Fig 5: whereas ALD outputs appear muted and blurred, PIDLD samples exhibit richer color fidelity and sharper detail. This visualization underscores PIDLD's superior performance in resource-constrained sampling regimes.

**Ablation Study.** We conducted a comparative analysis of PIDLD's performance under ablation settings involving the integral and differential terms, as shown in Fig 6. We observe that while both terms prove effective, the differential term is the primary contributor to quality improvement. This phenomenon results from the nature of annealed Langevin dynamics. The early-stage potential landscapes are smoothed by noise and exhibit shallow barriers between wells. In such scenarios, the integral term's ability to traverse local minima becomes less effective. In contrast, the differential term enables rapid convergence toward the centers of local wells at each noise scale, providing better initializations for subsequent sampling steps.

### 4.2 Solution Sampling for Reasoning Tasks

Given PIDLD's capacity to traverse local optima and sample from global energy minima, we evaluate the effectiveness of the proposed PIDLD when performing inference tasks using energy-based models. Such methods formulate reasoning problems as iterative energy minimization problems over learned energy functions that parameterize an energy landscape across all possible solutions. By applying PIDLD to the sampling process, we aim to address the limitations of conventional Langevin dynamics in navigating complex energy landscapes, thereby improving the overall performance and accuracy of energy models in solving reasoning tasks.

**Tasks and Base Model.** We follow existing work [6, 7] by selecting challenging reasoning tasks for our experiments. Specifically, we evaluate the performance of IRED [7] on Harder Datasets for Sudoku [38, 28] and Connectivity [5] problems. These Harder Datasets consist of out-of-distribution data relative to the training distribution, requiring models to utilize additional computational resources during inference to achieve effective generalization. In Sudoku, the model must complete a partially filled grid by predicting values that satisfy both Sudoku rules and the given constraints, with IRED using the SAT-Net [38] dataset (31-42 given numbers) for training, while evaluating generalization on the more challenging RRN dataset [28] (17-34 given numbers). The connectivity task requires a model to determine whether paths exist between node pairs in a graph given only the adjacency matrix, with IRED's training using graphs of up to 12 nodes, while the harder dataset challenges the model with larger 18-node graphs requiring more complex reasoning steps.

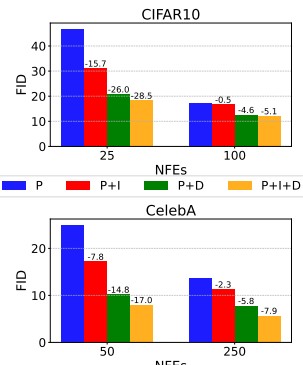

Figure 6: Ablation study of PIDLD under image sampling tasks. P+I+D denotes the complete model, while P+I, P+D, and P represent models with the derivative term, integral term, or both terms removed, respectively. Figures on the bar indicate performance improvements of each ablation compared to P.

**Baseline and Metrics.** We compare with the vanilla sampling method [7]. For both tasks, solution accuracy serves as the primary evaluation metric.

**Implementation.** In our experiments, we change the number of optimization steps in the original paper and finetune the hyperparameters to get the best accuracy. The step sizes are fixed and scheduled in the base model.

**Experiment Results.** The results are presented in Table 2, demonstrating that our proposed method consistently outperforms the baseline approach across varying NFEs on both Sudoku and Connectivity reasoning tasks. For the Sudoku task, our method achieves significant performance improvements ranging from 4.55% to 6.02% at low NFEs (5-15), while maintaining a stable advantage of approximately 3-5% at higher NFEs (30-80). Most notably, our approach reaches an accuracy of 50.54% with just 5 NFEs, surpassing the baseline's performance at 10 NFEs (51.00%), and demonstrating substantial computational efficiency.

For Connectivity, our method exhibits performance gains beginning at NFE=2 with improvements of 3.14%, 3.47%, 3.42%, 3.11%, and 3.27% across the 2-10 NFE range. Notably, our method achieves 91.32% accuracy at just NFE=2, already exceeding the baseline's maximum accuracy of 90.33% at NFE=10, representing a 5× improvement in computational efficiency. The overall results demonstrate both quality improvements and computational advantages over the baseline approach.

**Ablation Study.** We analyzed PIDLD performance through ablation studies of integral and derivative components (Fig. 7). Results indicate that combining both terms produces optimal performance across all sampling steps. The integral term provides consistent performance gains, while the derivative term primarily enhances early-stage convergence. As NFE increases, the derivative term's benefits diminish substantially, whereas the integral term's capacity to navigate local optima becomes dominant. This stems from the critical importance of attaining energy minima in reasoning tasks.

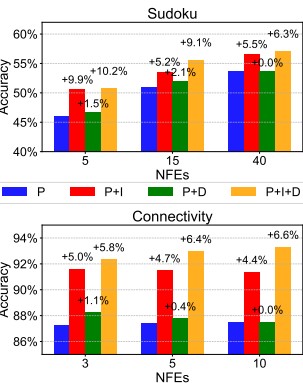

Figure 7: Ablation study of PIDLD under reasoning tasks. P+I+D denotes the complete model, while P+I, P+D, and P represent models with the derivative term, integral term, or both terms removed, respectively. Percentages indicate performance improvements of each ablation compared to P.

## 5 Discussion and Conclusion

We introduce PID-controlled Langevin Dynamics (PIDLD), a novel, training-free algorithm that accelerates sampling for generative models by reformulating the process through the lens of control theory. By leveraging an integral term for momentum and a derivative term for adaptive stabilization, PIDLD efficiently navigates complex energy landscapes to produce higher-quality samples in fewer steps. Its plug-and-play nature makes it immediately applicable to any pre-trained model reliant on Langevin sampling.

It is important to position PIDLD's contribution within the broader context of generative model acceleration. While significant progress has been made with methods like DDIM [32] and other ODE-based solvers [35], our work's primary focus is not to compete with these alternative generative frameworks. Instead, our goal is to fundamentally improve the foundational Langevin Dynamics sampler itself, which remains a cornerstone for many energy-based and score-based models. PIDLD is designed as a direct, drop-in enhancement for the standard LD sampling procedure within these models, whereas methods like DDIM alter the underlying diffusion process itself. The main application of our work is therefore to serve as a general and highly efficient inference accelerator for any pre-trained model that relies on Langevin-based sampling.

## Acknowledgments and Disclosure of Funding

This work is supported by Shenzhen Ubiquitous Data Enabling Key Lab under grant ZDSYS20220527171406015, the Tsinghua Shenzhen International Graduate School-Shenzhen Pengrui Endowed Professor-ship Scheme of Shenzhen Pengrui Foundation and the National Natural Science Foundation of China (No. U24B20180, 62476152).

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

# A    Comparison of Related Works

## A.1    Comparison of Langevin Sampling Speedup Techniques

Table 3: Comparison of Langevin Sampling Acceleration Methods

| Method | No Retrain[a] | No DtSts[b] | Theory Basis[c] | Simplicity[d] | Stability[e] |
|--------|-----------|---------|--------------|------------|-----------|
| Vanilla LD[1] | N/A | N/A | Base Sampling | Base | ✗ |
| CLD [3] | ✗ | ✓ | Statistic Mechanics | Low | ✓ |
| PDS [23, 24] | ✓ | ✗ | Matrix Preconditioning | Moderate | ✗ |
| Momentum-based [31, 40] | ✓ | ✓ | Optimization (Momentum) | High | ✗ |
| **PIDLD(Ours)** | ✓ | ✓ | Control Theory (PID) | High | ✓ |

[a] **No Retrain**: No retraining of the base generative model is required for acceleration.
[b] **No DtSts**: The core acceleration mechanism does not require dataset-specific statistics/priors.
[c] **Theory Basis**: The primary theoretical foundation of the acceleration method (or base method).
[d] **Simplicity**: Ease of modifying a standard Langevin sampler (or base method complexity).
[e] **Stability**: Employs an explicit or inherent mechanism to prevent common Langevin sampling instabilities (e.g., overshooting, oscillations).
[1] **Vanilla LD**: Standard Langevin Dynamics, serves as a baseline. "N/A" as it's not an acceleration method itself.

Table 3 provides a comparative overview of several methods for accelerating Langevin sampling, including our proposed PIDLD, against Vanilla Langevin Dynamics (LD) as a baseline. The comparison highlights key practical and theoretical characteristics: the necessity of model retraining, reliance on dataset-specific statistics for the core mechanism, underlying theoretical basis, integration simplicity, and critically, the employment of an explicit or inherent mechanism for instability prevention (e.g., against overshooting or oscillations).

Our method, PIDLD, stands out by achieving high simplicity and flexibility, requiring no model retraining and operating independently of dataset-specific statistics, similar to Momentum-based approaches. However, PIDLD uniquely incorporates a control-theoretic (PID) framework that provides an explicit mechanism for instability prevention through its derivative (D) term, which adaptively damps updates based on gradient trends. This direct approach to stability contrasts with Momentum-based methods, which lack such an adaptive damping feature and are thus marked as not explicitly preventing these common Langevin instabilities. CLD also inherently prevents instability through its critically-damped formulation within a retrained, augmented space. PDS, while not requiring retraining, relies on dataset statistics for optimal performance and does not feature an adaptive instability prevention mechanism in its sampling dynamics. Therefore, PIDLD offers a compelling combination of ease of use, broad applicability, and enhanced stability through its novel control-inspired design.

## A.2    Positioning PIDLD Among Sampling Acceleration Strategies

Beyond the momentum-based approaches discussed in the main text, the field of accelerating generative samplers encompasses several distinct strategies. A notable alternative paradigm involves learning an entirely new, often near-optimal, sampling process, rather than modifying an existing one. Methods grounded in Stochastic Optimal Control (SOC), for instance [4, 12], exemplify this approach. They aim to learn a novel drift function that transports a base distribution to the target distribution, a process that typically requires a full training or fine-tuning phase to optimize a specific objective. In contrast, PIDLD is designed not to replace the sampling process, but to enhance it as a training-free, inference-time accelerator. Instead of learning a new policy, PIDLD applies principles from classical control theory to improve the numerical stability and convergence speed of the existing Langevin dynamics update rule. This positions PIDLD as a general-purpose and readily applicable tool for boosting the efficiency of any pre-trained model reliant on Langevin sampling, complementing approaches that require dedicated training.

## B  Proof of Proposition 1

**Lemma 1** (Convergence of Covariance Series). *Let $J \in \mathbb{R}^{n \times n}$ with spectral radius $\rho(J) < 1$, and $Q \succeq 0$. Then the series $S = \sum_{k=0}^{\infty} J^k Q (J^k)^T$ converges.*

*Proof.* (Proof of Lemma 1) Fix an induced matrix norm. By Gelfand's formula, for any $\delta > 0$, there exists $C > 0$ such that $\|J^k\| \leq C(\rho(J) + \delta)^k$ for all $k \geq 0$. Choose $\delta = (1 - \rho(J))/2$, ensuring $\rho(J) + \delta < 1$. Applying submultiplicativity:

$$\|J^k Q (J^k)^T\| \leq \|J^k\|^2 \|Q\| \leq C^2 \|Q\| (\rho(J) + \delta)^{2k}.$$

The series converges absolutely by comparison with the geometric series $\sum_{k=0}^{\infty} (\rho(J) + \delta)^{2k}$. $\quad\square$

*Proof.* (Proof of Proposition 1) **Deterministic Stability:** Define the state vector $v_t = \begin{bmatrix} x_t \\ x_{t-1} \end{bmatrix}$.

Linearizing around $x^*$, let $\delta_t = v_t - \begin{bmatrix} x^* \\ x^* \end{bmatrix}$. Using $\nabla U(x) \approx -m(x - x^*)$, the dynamics become:

$$\delta_{t+1} = J \delta_t, \quad J = \begin{bmatrix} 1 - \epsilon(1 + k_d)m & \epsilon k_d m \\ 1 & 0 \end{bmatrix}.$$

The characteristic equation $\det(\lambda I - J) = \lambda^2 - \lambda(1 - \epsilon(1 + k_d)m) + \epsilon k_d m = 0$. Applying Jury's stability criteria:

1. $|\epsilon k_d m| < 1$

2. $1 - \epsilon(1 + k_d)m + \epsilon k_d m > 0 \Rightarrow \epsilon m > 0$

3. $1 + \epsilon(1 + k_d)m + \epsilon k_d m > 0 \Rightarrow 2 - \epsilon(1 + 2k_d)m > 0 \Rightarrow \epsilon < \frac{2}{(1 + 2k_d)m}$

The above leads to $\epsilon < \frac{2}{(1 + 2k_d)m}$.

**Stochastic Case:** The linearized system becomes:

$$\delta_{t+1} = J \delta_t + w_t, \quad w_t = \begin{bmatrix} \sqrt{2\epsilon}\xi_t \\ 0 \end{bmatrix} \sim \mathcal{N}(0, Q), \ Q = \begin{bmatrix} 2\epsilon I & 0 \\ 0 & 0 \end{bmatrix}.$$

The stationary distribution covariance can be computed: $\Sigma_\infty = \sum_{k=0}^{\infty} J^k Q (J^k)^T$. By Lemma 1 with $\rho(J) < 1$ (guaranteed by $\epsilon < 2/[(1 + 2k_d)m]$), the series converges. Hence $\delta_t$ converges to $\mathcal{N}(0, \Sigma_\infty)$. $\quad\square$

Note that the m-strong convexity is a standard assumption for local stability analysis and does not presume the global energy landscape is convex. The assumption in Proposition 1 is intentionally narrow and serves a specific purpose: to provide a controlled, tractable setting to formally analyze the local behavior of our sampler. Its goal is to prove that our novel derivative (D) term provides the intended stabilizing effect near a potential minimum, without disrupting the fundamental convergence properties of Langevin dynamics [11, 29]. It provides a theoretical justification for the stability enhancements we observe empirically.

## C  Detailed Experiment Settings

### C.1  Toy Experiment Settings

Our toy experiment generally follows the setting of NCSN [33]. We choose $p_{\text{data}} = \frac{1}{5}\mathcal{N}((-5, -5), I) + \frac{4}{5}\mathcal{N}((5, 5), I)$. The score network is a 3-layer MLP with 128 hidden units and softplus activation functions. We train the score network with anneal denoising score matching for 60 epochs on 100000 training samples with a batch size of 128. We use Adam optimizer with a learning rate of 0.0001, $\beta_1 = 0.9$ and $\beta_2 = 0.999$.

For the sampling process, we choose 1280 initial samples uniformly in the square $[-8, 8] \times [-8, 8]$. We use annealed Langevin dynamics where $L = 8$, $T = 150$ and $\epsilon = 8 \times 10^{-6}$. We choose $\{\sigma_i\}_{i=1}^{L}$ to be a geometric progression, with $\sigma_1 = 20$ and $\sigma_8 = 0.01$. We use the learned score function for sampling.

## C.2 Detailed Experiment Implementation of Image Generation

Table 4: Hyperparameter settings of PIDLD and ALD in image generation.

| Dataset | SGM | | | | | EBM | | | | |
|---|---|---|---|---|---|---|---|---|---|---|
| | | PIDLD | | | ALD | | PIDLD | | | ALD |
| | NFEs | $k_p$ | $k_i$ | $k_d$ | $k_p$ | NFEs | $k_p$ | $k_i$ | $k_d$ | $k_p$ |
| | 232×5 | 0.90 | 0.00 | 1.00 | 0.90 | 40 | 0.50 | 0.80 | 1.50 | 0.70 |
| | 232×3 | 1.50 | 0.00 | 1.20 | 1.00 | 30 | 0.80 | 0.20 | 2.00 | 1.20 |
| CIFAR10 | 232×1 | 2.00 | 0.00 | 3.00 | 1.75 | 20 | 1.20 | 0.50 | 2.50 | 2.00 |
| | 100×1 | 2.00 | 0.50 | 4.50 | 2.50 | 10 | 1.50 | 1.00 | 3.50 | 3.00 |
| | 25×1 | 3.75 | 0.25 | 4.00 | 6.00 | – | – | – | – | – |
| | 500×5 | 1.00 | 0.00 | 3.00 | 0.90 | 30 | 0.60 | 0.50 | 2.50 | 0.80 |
| | 500×3 | 1.25 | 0.00 | 3.00 | 1.25 | 25 | 0.90 | 0.30 | 3.50 | 1.50 |
| CelebA | 500×1 | 2.00 | 0.00 | 3.00 | 2.00 | 20 | 1.50 | 0.80 | 4.00 | 2.50 |
| | 250×1 | 2.00 | 0.50 | 3.00 | 2.50 | 15 | 1.80 | 1.20 | 5.00 | 3.50 |
| | 50×1 | 3.75 | 1.00 | 3.75 | 5.50 | – | – | – | – | – |

For SGM, we use the pretrained checkpoint provided by the official repository of NCSNv2 [34] to initialize the weights of the score network. We also apply the exponential moving average trick to the weights. The score network uses the RefineNet [21] architecture. For details please refer to the original paper [34]. For EBM, we use the pretrained checkpoint provided by the official repository of IGEBM [8] for CIFAR10 dataset, and retrain the model for the CelebA dataset.

For the sampling process, we follow the setting of NCSNv2 to use $\sigma_1 = 50.0$, $\sigma_L = 0.01$, $L = 232$, $T = 5$, $\epsilon = 6.2 \times 10^{-6}$ for CIFAR10 and $\sigma_1 = 90.0$, $\sigma_L = 0.01$, $L = 500$, $T = 5$, $\epsilon = 3.3 \times 10^{-6}$ for CelebA. We also follow NCSNv2 to add an additional denoising step to the last sample, given by $x_{\text{final,denoised}} = x_{\text{final}} + \sigma_L^2 s_\theta(x_{\text{final}}, \sigma_L)$. For IGEBM, the sampling step corresponds to the NFE. We decay $k_i$ at each step. For the step size, we use the default $\epsilon = 100$ for CIFAR10 and $\epsilon = 400$ for CelebA. We keep $\epsilon$ fixed across all experiments.

We generate 10000 images in each run and use FID to evaluate the generation quality. We input the raw generated images to the inception network without clipping the pixel values to $[0, 1]$.

Since conventional ALD is the degenerate case of PIDLD where $k_i = k_d = 0$, we tune only $k_p$ to get the best performance of vanilla ALD and tune $k_p, k_i, k_d$ to optimize PIDLD. We give our hyperparameter settings in Table 4.

For convenience in parameter tuning, we tune only $k_p$ and $k_d$ when the number of sampling steps is high, because the derivative term is the dominant factor in anneal Langevin dynamics sampling, as mentioned in the ablation study. We observe that when NFEs decrease, the optimal $k_p, k_i, k_d$ show an increasing trend. This is intuitive since fewer steps necessitate more updates per step.

## C.3 Detailed Experiment Implementation of Reasoning Tasks

We apply the original repository of IRED [7] to implement PIDLD. We use the pretrained checkpoint provided. For the Sudoku task, the energy network uses ResNet as the backbone. For connectivity task, the energy networks uses Graph Neural Network (GNN) as the backbone. Please refer to the original paper for model details.

For both tasks, we sample 1000 labeled items for evaluation in the harder dataset, following the original setting of IRED. We give our hyperparameter settings in Table 5.

Table 5: Hyperparameter settings of PIDLD and IRED sampling in reasoning tasks.

| Dataset | Method | **PIDLD** | | | | **IRED** |
|---|---|---|---|---|---|---|
| | | NFEs | $k_p$ | $k_i$ | $k_d$ | $k_p$ |
| Sudoku | | 5 | 1.20 | 4.00 | 0.50 | 1.50 |
| | | 10 | 1.00 | 4.00 | 0.50 | 1.50 |
| | | 15 | 0.90 | 4.00 | 0.50 | 1.00 |
| | | 30 | 0.80 | 4.00 | 0.50 | 1.00 |
| | | 40 | 0.80 | 3.00 | 0.50 | 0.80 |
| | | 80 | 0.50 | 3.00 | 0.50 | 0.80 |
| | | NFEs | $k_p$ | $k_i$ | $k_d$ | $k_p$ |
| Connectivity | | 5 | 1.10 | 0.40 | 0.01 | 1.50 |
| | | 10 | 1.00 | 0.40 | 0.01 | 1.50 |
| | | 15 | 0.90 | 0.40 | 0.01 | 1.50 |
| | | 30 | 0.80 | 0.30 | 0.01 | 1.00 |
| | | 40 | 0.80 | 0.30 | 0.01 | 1.00 |
| | | 80 | 0.80 | 0.20 | 0.01 | 1.00 |

# D  Additional Experiments on the Effect of Integral and Derivative Terms

## D.1  Effect Validation on Integral Term

To validate the effect of integral term on mitigating bias, we provide more empirical evidence by running toy experiments on two-dimensional points (see Appendix C.1 for experiment settings) with only $P$ term, with $P, I$ terms, and with $P, D$ terms. To simulate a bias in gradient estimation, we add a constant perturbation to the score that points to the direction of $(-1, 1)$ and is scaled by $1/20$ of the Frobenius norm of the gradient. We use $d$, the Euclidean distance of the final cluster center to the true center, as the metric of bias. Here the subscript 1 and 2 corresponds to the cluster near $(5, 5)$ and $(-5, -5)$, respectively. The results are shown in Table 6 and Table 7. We run 100 experiments with different random seeds under each hyperparameter setting. The number before and after $\pm$ represents mean and standard deviation, respectively.

Table 6: Effect of $k_i$ on bias

| $k_i$ | $d_1$ | $d_2$ |
|---|---|---|
| 0.00 | $0.0463 \pm 0.0237$ | $0.2729 \pm 0.0840$ |
| 0.05 | $0.0414 \pm 0.0235$ | $0.2711 \pm 0.0816$ |
| 0.10 | $0.0379 \pm 0.0215$ | $0.2559 \pm 0.0802$ |
| 0.15 | $0.0339 \pm 0.0190$ | $0.2485 \pm 0.0781$ |
| 0.20 | $0.0326 \pm 0.0192$ | $0.2436 \pm 0.0756$ |
| 0.25 | $0.0317 \pm 0.0176$ | $0.2356 \pm 0.0742$ |
| 0.30 | $0.0315 \pm 0.0184$ | $0.2307 \pm 0.0714$ |
| 0.35 | $0.0308 \pm 0.0169$ | $0.2330 \pm 0.0708$ |

Table 7: Effect of $k_d$ on bias

| $k_d$ | $d_1$ | $d_2$ |
|---|---|---|
| 0.0 | $0.0463 \pm 0.0237$ | $0.2729 \pm 0.0840$ |
| 2.0 | $0.0445 \pm 0.0223$ | $0.2904 \pm 0.0882$ |
| 4.0 | $0.0448 \pm 0.0226$ | $0.2971 \pm 0.0909$ |
| 6.0 | $0.0453 \pm 0.0218$ | $0.3090 \pm 0.0914$ |
| 8.0 | $0.0456 \pm 0.0213$ | $0.3250 \pm 0.0871$ |
| 10.0 | $0.0455 \pm 0.0214$ | $0.3387 \pm 0.0914$ |
| 12.0 | $0.0466 \pm 0.0204$ | $0.3469 \pm 0.0873$ |
| 14.0 | $0.0522 \pm 0.0230$ | $0.3431 \pm 0.0911$ |

It can be observed that increasing $k_i$ helps to decrease $d_1$ and $d_2$, the final bias of cluster centers, while increasing $k_d$ won't yield the same effect. This further demonstrates that the integral term makes the sampling algorithm robust to gradient bias.

## D.2  Effect Validation on Derivative Term

To validate the effect of derivative term on reducing oscillation, we provide more empirical evidence by running toy experiments on two-dimensional points (see Appendix C.1 for experiment settings) with only $P$ term, with $P, I$ terms, and with $P, D$ terms.

It is hard to measure the instability of the trajectory of a single point, because the Langevin dynamics itself adds noise at each step, and the trajectory is intrinsically stochastic. But the randomness can be

reduced by choosing a group of points. Since the generated data roughly follows a GMM distribution, we use the centers estimated by Gaussian mixture model as representative objects, and study the trajectories of the two centers.

We use 8 noise levels and 150 steps for each level ($L = 8, T = 150$). We record the positions of the two centers every 5 steps, so there are 240 recorded time steps in total. In the earlier stage the two clusters does not separate and the mean estimation is not reliable, so we use the last 200 steps.

To quantify the amplitude of oscillation of a trajectory, we use the averaged Euclidean distance to the final position, $d_{\text{sum}} = \sum_{i=41}^{240} \sqrt{(x_i - x_{\text{final}})^2 + (y_i - y_{\text{final}})^2}$, as the major indicator. We also report the maximum distance, $d_{\text{max}}$, which can be seen as a measure of peak overshoot. Borrowing from control theory, we also add the settling time metric, which is defined as the time index where $\|p_t^{(j)} - p_{\text{final}}^{(j)}\|$ of all subsequent time steps are less than 0.1. Here $p_t^{(j)}$ is defined as the position of cluster center at time index $t$, and $p_{\text{final}}^{(j)}$ is defined as the final position of the cluster center. The superscript $j = 1, 2$ indicates the center near $(5, 5)$ and $(-5, -5)$, respectively.

We fix $k_p = 1.5$ to boost oscillation, and change $k_i$ and $k_d$. We run 100 experiments with different random seeds under each hyperparameter setting. The number before and after $\pm$ represents mean and standard deviation, respectively. The results are shown in the following tables.

Table 8: Metrics for Cluster 1 with varying $k_d$

| $k_d$ | $d_{\text{sum}}^{(1)}$ | $d_{\text{max}}^{(1)}$ | $t_{\text{settling}}^{(1)}$ |
|---|---|---|---|
| 0.0 | $34.97 \pm 3.00$ | $1.61 \pm 0.16$ | $107.05 \pm 12.43$ |
| 2.0 | $31.02 \pm 2.67$ | $1.35 \pm 0.13$ | $105.26 \pm 12.30$ |
| 4.0 | $28.76 \pm 2.67$ | $1.25 \pm 0.52$ | $105.10 \pm 12.76$ |
| 6.0 | $27.00 \pm 2.41$ | $1.11 \pm 0.10$ | $104.63 \pm 12.70$ |
| 8.0 | $25.79 \pm 2.37$ | $1.08 \pm 0.11$ | $103.35 \pm 12.86$ |
| 10.0 | $26.40 \pm 2.61$ | $1.18 \pm 0.13$ | $103.37 \pm 13.67$ |
| 12.0 | $34.54 \pm 4.56$ | $3.69 \pm 2.01$ | $102.44 \pm 12.97$ |

Table 9: Metrics for Cluster 2 with varying $k_d$

| $k_d$ | $d_{\text{sum}}^{(2)}$ | $d_{\text{max}}^{(2)}$ | $t_{\text{settling}}^{(2)}$ |
|---|---|---|---|
| 0.0 | $69.30 \pm 7.81$ | $3.11 \pm 0.72$ | $145.88 \pm 12.16$ |
| 2.0 | $48.74 \pm 7.20$ | $2.17 \pm 1.11$ | $144.30 \pm 12.67$ |
| 4.0 | $39.42 \pm 5.48$ | $1.63 \pm 1.53$ | $144.13 \pm 11.27$ |
| 6.0 | $35.75 \pm 4.65$ | $1.30 \pm 0.28$ | $143.29 \pm 13.69$ |
| 8.0 | $33.22 \pm 4.12$ | $1.25 \pm 0.22$ | $142.09 \pm 13.38$ |
| 10.0 | $36.20 \pm 4.66$ | $1.73 \pm 0.35$ | $141.42 \pm 12.55$ |
| 12.0 | $70.52 \pm 12.14$ | $8.45 \pm 4.46$ | $139.96 \pm 12.60$ |

Table 10: Metrics for Cluster 1 with varying $k_i$

| $k_i$ | $d_{\text{sum}}^{(1)}$ | $d_{\text{max}}^{(1)}$ | $t_{\text{settling}}^{(1)}$ |
|---|---|---|---|
| 0.00 | $34.97 \pm 3.00$ | $1.61 \pm 0.16$ | $107.05 \pm 12.43$ |
| 0.05 | $34.77 \pm 2.92$ | $1.60 \pm 0.17$ | $106.67 \pm 12.42$ |
| 0.10 | $34.65 \pm 2.92$ | $1.60 \pm 0.18$ | $106.73 \pm 12.61$ |
| 0.15 | $34.46 \pm 2.88$ | $1.60 \pm 0.18$ | $105.96 \pm 12.80$ |
| 0.20 | $34.35 \pm 2.82$ | $1.59 \pm 0.17$ | $106.18 \pm 12.10$ |
| 0.25 | $34.24 \pm 2.84$ | $1.62 \pm 0.37$ | $106.59 \pm 12.61$ |
| 0.30 | $34.09 \pm 2.80$ | $1.58 \pm 0.18$ | $106.26 \pm 12.33$ |
| 0.35 | $34.05 \pm 2.85$ | $1.63 \pm 0.56$ | $106.29 \pm 12.86$ |
| 0.40 | $33.88 \pm 2.74$ | $1.58 \pm 0.18$ | $106.11 \pm 12.91$ |

Table 11: Metrics for Cluster 2 with varying $k_i$

| $k_i$ | $d_{\text{sum}}^{(2)}$ | $d_{\text{max}}^{(2)}$ | $t_{\text{settling}}^{(2)}$ |
|---|---|---|---|
| 0.00 | $69.30 \pm 7.81$ | $3.11 \pm 0.72$ | $145.88 \pm 12.16$ |
| 0.05 | $70.95 \pm 8.19$ | $3.09 \pm 0.67$ | $146.24 \pm 12.57$ |
| 0.10 | $72.61 \pm 8.41$ | $3.15 \pm 0.75$ | $147.85 \pm 11.44$ |
| 0.15 | $74.00 \pm 8.39$ | $3.19 \pm 0.73$ | $147.20 \pm 11.00$ |
| 0.20 | $75.67 \pm 7.84$ | $3.20 \pm 0.65$ | $148.30 \pm 11.94$ |
| 0.25 | $77.05 \pm 8.64$ | $3.30 \pm 1.04$ | $148.29 \pm 10.75$ |
| 0.30 | $78.40 \pm 8.38$ | $3.30 \pm 0.76$ | $148.50 \pm 11.69$ |
| 0.35 | $79.88 \pm 8.59$ | $3.39 \pm 1.31$ | $148.46 \pm 10.75$ |
| 0.40 | $80.77 \pm 8.16$ | $3.34 \pm 0.75$ | $147.77 \pm 10.30$ |

It can be seen that for each cluster, the metrics decrease as $k_d$ increases, while they do not show comparable decreasing trend as we increase $k_i$. The results further validate our claim that compared to the integral term, the derivative term has the advantage on reducing the oscillation in the sampling process.

## D.3 Effect Validation on Integral Term Decay

We use SGM image generation task to further validate the effect of decaying integral term coefficient. For CIFAR10, we generate images with 100 noise levels and 1 step per level (NFE=$100 \times 1$). We use $k_p = 2.0$, $k_d = 4.5$, and change $k_i$ and $\gamma$. For CelebA, we generate images with 50 noise levels and 1 step per level (NFE=$50 \times 1$). We use $k_p = 3.75$, $k_d = 3.75$, and change $k_i$ and $\gamma$. The results are shown in Table 12 and Table 13.

Table 12: CIFAR10 FID of $100 \times 1$ sampling steps ($k_p = 2.0$, $k_d = 4.5$). Bold: best in column.

| $\gamma \setminus k_i$ | 0.250 | 0.375 | 0.500 | 0.625 | 0.750 | 0.875 |
|---|---|---|---|---|---|---|
| 1 | **12.29** | 12.28 | 12.32 | 12.41 | 12.53 | 12.69 |
| 0.99 | 12.31 | **12.22** | **12.21** | 12.20 | 12.24 | 12.27 |
| 0.98 | 12.31 | 12.28 | 12.23 | **12.20** | **12.16** | **12.15** |
| 0.97 | 12.36 | 12.28 | 12.25 | 12.24 | 12.20 | 12.18 |
| 0.96 | 12.41 | 12.33 | 12.28 | 12.25 | 12.23 | 12.22 |

Table 13: CelebA FID of $50 \times 1$ sampling steps ($k_p = 3.75$, $k_d = 3.75$). Bold: best in column.

| $\gamma \setminus k_i$ | 0.750 | 0.875 | 1.000 | 1.125 | 1.250 | 1.375 |
|---|---|---|---|---|---|---|
| 1 | **8.55** | 8.08 | 7.97 | 8.47 | 9.02 | 9.56 |
| 0.99 | 10.67 | **8.01** | **7.53** | 7.76 | 8.13 | 8.55 |
| 0.98 | 16.52 | 11.24 | 8.59 | **7.71** | **7.65** | 7.86 |
| 0.97 | 23.17 | 16.87 | 12.28 | 9.53 | 8.21 | **7.75** |
| 0.96 | 29.20 | 23.14 | 17.84 | 13.71 | 10.86 | 9.14 |

We find that:

1. The best performance of tuning both $k_i$ and $\gamma$ is better than the best performance of tuning only $k_i$;

2. The higher the $k_i$ is, the higher decay (lower $\gamma$) should be applied to balance the effect of larger integral gain, which aligns with common intuition.

The result is a strong evidence that the decay in integral term does improve sampling performance.

# E  Computation Cost Comparison

We show that our approach to measure computation cost by NFE is reasonable by conducting experiments to compare the physical time consumption. For (non-degenerate) PIDLD, we use the original code. For vanilla ALD, we modify our code implementation by deleting all PID related terms in the sample updating function, which ensures no additional computation cost by PIDLD is introduced. We run both algorithms on NVIDIA A800-SXM4-40GB. The results are shown in Table 14.

Table 14: Physical time comparison of vanilla ALD and PIDLD.

| Model | Dataset | NFE | Time Consumed |
|---|---|---|---|
| PIDLD | CIFAR10 | $100 \times 1$ | 7min54s |
| Vanilla ALD | CIFAR10 | $100 \times 1$ | 7min43s |
| PIDLD | CelebA | $50 \times 1$ | 13min07s |
| Vanilla ALD | CelebA | $50 \times 1$ | 12min52s |

It can be seen that there is no significant difference in the time cost between PIDLD and vanilla ALD. For other numbers of NFE, the computation time changes proportionally with NFE, and the time consumption of vanilla ALD and PIDLD is still close. The result is actually intuitive, since the main computation cost is on the forward propagation of the score network, which involves a large amount of matrix multiplications. Adding PID terms just brings some matrix addition operations, so there wouldn't be much increase in the computation time, and thus NFE is an accurate measure of real computing time.

# F  Inception Score Statistics

For CV task, apart from FID, we also report inception score [30] statistics of score based generative models on CIFAR10 dataset in Table 15.

Table 15: Inception score statistics of SGM on CIFAR10 dataset.

| NFE | ALD | PIDLD |
|---|---|---|
| $232 \times 5$ | $8.45 \pm 0.18$ | $8.43 \pm 0.11$ |
| $232 \times 3$ | $8.47 \pm 0.25$ | $8.58 \pm 0.21$ |
| $232 \times 1$ | $7.92 \pm 0.22$ | $8.46 \pm 0.22$ |
| $100 \times 1$ | $7.77 \pm 0.25$ | $8.26 \pm 0.26$ |
| $25 \ \times 1$ | $7.07 \pm 0.24$ | $7.65 \pm 0.21$ |

In terms of inception score, we find that PIDLD still generally outperforms ALD under different NFEs, which serves as another evidence of its advantage.

As for CelebA dataset, the inception score metric is not applicable, because the inception model is trained on ImageNet dataset which consists of natural objects, while CelebA is a human face dataset. The distribution shift is large, so it is not reasonable to apply ImageNet inception feature extractor. It is actually a common practice to omit inception score for CelebA.

# G  Additional Examples of Generated Images

We provide more image samples generated by SGM using PIDLD as in Fig 8 and Fig. 9.

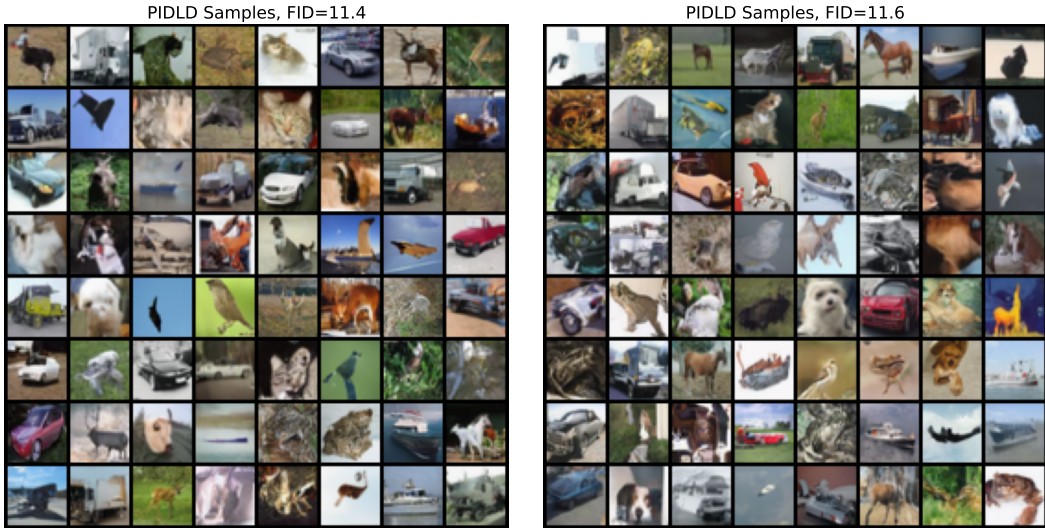

(a) CIFAR10 samples with 232×5 sampling steps.      (b) CIFAR10 samples with 232×1 sampling steps.

Figure 8: Visualization of CIFAR10 images generated by SGM with PIDLD.

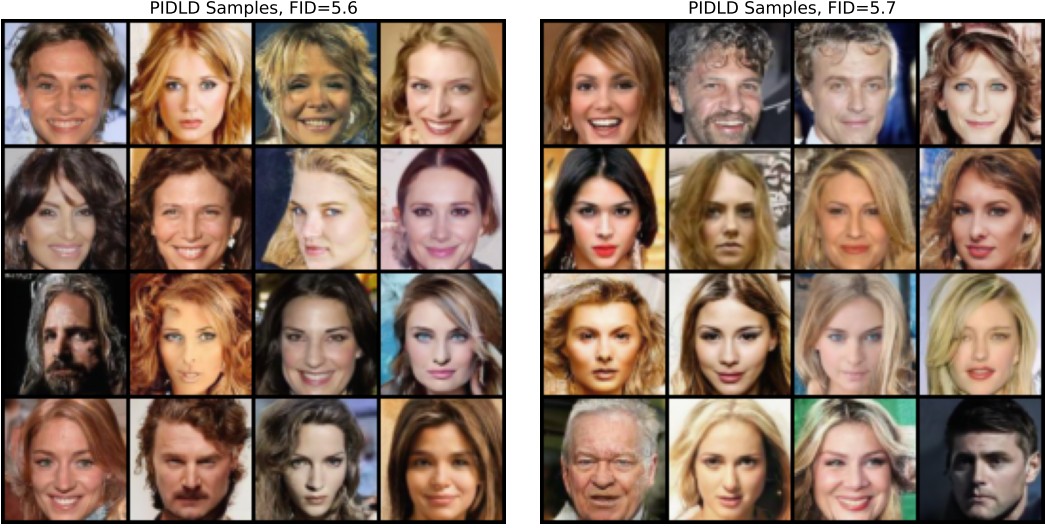

(a) CelebA samples with 500×5 sampling steps.      (b) CelebA samples with 500×1 sampling steps.

Figure 9: Visualization of CelebA images generated by SGM with PIDLD.

## H    Limitations and Future Work

PIDLD presents a promising approach to accelerate Langevin-based sampling, and we identify several avenues for future development. Firstly, while our parameter analysis offers guidance, exploring adaptive or automated strategies for tuning the PID coefficients $(k_p, k_i, k_d, \gamma)$ could further enhance its plug-and-play capability across diverse models and datasets. Secondly, building upon our initial stability analysis (Proposition 1), a deeper theoretical investigation into the convergence guarantees and acceleration properties of the full PID controller, especially in the context of non-convex landscapes characteristic of EBMs [8] and SGMs [33], would be valuable. Lastly, our proposed method may be beneficial in extending the PID control principles to more general ODE and SDE solvers, which are prevalent in recent score-based models [35]. This represents an exciting direction to broaden the applicability of PIDLD and potentially unlock further performance gains.

