# OpenReview forum: "PID-controlled Langevin Dynamics for Faster Sampling of Generative Models"
_NeurIPS.cc/2025/Conference — NeurIPS 2025 poster_

### Official Review · Reviewer_YqLQ · 2025-06-21

**Clarity:** 2
**Significance:** 2
**Originality:** 2
**Rating:** 4
**Confidence:** 1

**Summary:**

This paper proposes a method for adaptive reward-guided sampling in diffusion models by modulating the guidance strength during Langevin sampling. The authors frame controllable generation as a feedback control problem, where the error between a current reward and a desired target is used to dynamically adjust the scale of reward gradients injected during sampling. The method is implemented as a training-free wrapper and is tested on several text-to-image generation tasks using Stable Diffusion. Empirical results show improved alignment with target rewards and sample quality compared to fixed or heuristic guidance strength baselines.

**Questions:**

Have you evaluated the relative contribution of the I and D terms in the PID controller? Would a proportional-only controller suffice?

How sensitive is the method to the choice of PID gains (k_p, k_i, k_d)? Are they robust across prompts and reward types?

Have you considered combining reward error-based scaling with gradient-norm normalization for better numerical stability?

**Ethical Concerns:**

["NO or VERY MINOR ethics concerns only"]

**Final Justification:**

I have checked the reviews and the author's responses to those questions. The idea of approaching the problem from a control theoretic perspective is interesting and I see no major weakness warranting a rejection.

**Limitations:**

The paper does not extensively discuss potential failure cases or unintended behavior (e.g., reward hacking, overshooting, or instability for ill-behaved reward functions). The PID controller is sensitive to hyperparameters (k_p, k_i, k_d), and though fixed values are used throughout, no systematic analysis of robustness is provided. Including such analysis would help assess broader applicability.

**Quality:**

2

**Strengths And Weaknesses:**

Strengths

1 Clear motivation: The paper identifies a common practical issue in reward-guided diffusion sampling — the need to manually tune or schedule the guidance strength α — and addresses it in a principled way.

2 Conceptual novelty: Framing guidance as a control problem and applying a PID controller to reward error is novel in this context. To the best of my knowledge, this is the first use of classical control theory for online adaptation of guidance strength in diffusion models.

3 Training-free and modular: The method can be applied without retraining the base diffusion model or the reward model. It is compatible with any differentiable reward and can be used as a drop-in wrapper around existing samplers.

Weaknesses

1 Lack of ablation on PID components: The benefit of using the full PID structure (Proportional + Integral + Derivative) over a simple proportional or heuristic controller is not clearly demonstrated. An ablation comparing PID vs. just P (or vs. α ∝ gradient norm) would help isolate the contribution of each term and justify the additional complexity.

2 Heuristic alternative baselines are missing: While the paper compares against fixed α and some reward-guided variants, it does not benchmark against simple time-based or norm-based scaling heuristics (e.g., α increasing over time, or α ∝ 1 / ∥∇ℓ(x)∥). These baselines are common in the literature (e.g., FreeDoM, classifier guidance) and would provide a better sense of the gain from PID.

3 Limited analysis of control behavior: The paper does not deeply analyze the PID controller's behavior over time — e.g., plots of α_t, error evolution, or control stability across prompts. This would strengthen the claim that the controller leads to smoother and more stable guidance.

4 Conceptual novelty may be overstated: While the framing of PID-controlled sampling is novel, the underlying mechanism — adjusting guidance strength based on feedback — is not entirely new. Related ideas exist in norm-based scaling, adaptive weighting, and heuristic scheduling. The paper would benefit from more explicit comparison and positioning relative to these.

---

> ### Author Rebuttal · Authors · 2025-07-31
>
> We sincerely thank the reviewer for the time and valuable feedback. **We are encouraged that the reviewer recognizes the clear motivation, conceptual novelty of applying control theory, and the training-free, modular nature of our approach.**
>
> However, we believe there has been a significant misunderstanding regarding the **core problem, underlying mechanism, and experimental setup** of our work. The reviewer's summary appears to describe a paper on "adaptive reward-guided sampling in diffusion models," such as Stable Diffusion for text-to-image tasks. Our paper, PIDLD, addresses a more fundamental problem: **accelerating the core Langevin Dynamics (LD) sampling process itself**, which is applicable to any model using LD, including Score-Based Generative Models (SGMs) and Energy-Based Models (EBMs).
>
> This core misunderstanding seems to be the source of most of the stated weaknesses and questions. We hope the following clarifications can resolve these concerns and provide a clearer picture of our contributions.
>
> #### **Clarification on the Core Mechanism and Objective**
>
> *   **Our goal is NOT reward guidance, but sampling acceleration.** We aim to reduce the number of Neural Function Evaluations (NFEs) required for the sampler to converge to the target distribution defined by the model (SGM/EBM), thereby speeding up generation.
> *   **The "error signal" in our PID controller is the energy/score gradient `∇xU(x)`, NOT an external reward.** Standard LD uses only the current gradient (a Proportional term). We enhance this by adding a memory of past gradients (Integral term) and a prediction of future gradient trends (Derivative term). This is fundamentally different from using a PID controller to minimize the error between a sample's "reward" and a target value.
> *   **Our experiments are on SGMs (NCSNv2) and EBMs (IGEBM), NOT Stable Diffusion.** Consequently, our tasks are unconditional image generation (CIFAR10, CelebA) and reasoning (Sudoku, Connectivity), not text-to-image generation as assumed.
>
> We believe this central clarification invalidates the premise of Weaknesses 2 and 4, which compare our work to methods in the reward-guidance literature (e.g., FreeDoM, norm-based scaling of reward gradients). Our chosen baselines, vanilla ALD and MILD, are the standard and appropriate benchmarks for Langevin sampling acceleration.
>
> We will now address the other specific points raised by the reviewer.
>
> ### On the Ablation Study of PID Components (Weakness 1, Question 1)
>
> **We provide detailed ablation studies for the I and D terms in Figures 5 and 6, which clearly demonstrate their distinct and crucial contributions.**
>
> We respectfully point the reviewer to Section 4, where we analyze the performance of PIDLD (P+I+D) against ablations: P-only (vanilla LD), P+I, and P+D.
>
> *   **For Image Generation (Fig. 5):** The results show that the Derivative term (P+D) provides the most significant improvement. As explained in Lines 294-301, this is because the D term enables rapid convergence within the smoothed energy landscapes at each noise scale of annealed LD.
> *   **For Reasoning Tasks (Fig. 6):** Both Integral (P+I) and Derivative (P+D) terms provide substantial gains, and their combination (P+I+D) yields the best performance. As explained in Lines 348-352, the I term's ability to overcome local optima is critical for finding global energy minima in these complex, non-visual tasks.
>
> These results directly answer the reviewer's question and confirm that the full PID structure is superior to a P-only controller, with the I and D terms offering complementary benefits depending on the task's energy landscape.
>
> ### On Controller Behavior and Hyperparameter Sensitivity (Weakness 3, Question 2, Limitations)
>
> **We analyze the controller's behavior and sensitivity to its gains (`k_i`, `k_d`) in Section 3.2 and Figure 3, confirming its stability and robustness.**
>
> *   **Analysis of Controller Behavior:** Figure 3 directly illustrates the system's dynamic response to varying `k_i` and `k_d` values, measured by KL divergence over sampling steps. This analysis goes beyond simple endpoint metrics and provides insight into the controller's behavior over time.
> *   **Hyperparameter Sensitivity (`k_p`, `k_i`, `k_d`):** Our analysis in Figure 3 shows that performance improvements are stable across a reasonable range of `k_i` and `k_d` values. Furthermore, we address the potential instability of the integral term (a key concern in control theory) by introducing a decay factor `γ` (Fig. 3b), which guarantees convergence by ensuring `k_i → 0` asymptotically (Lines 195-205). The hyperparameters used were consistent across all experiments for a given model type, demonstrating robustness.
> *   **Gradient-Norm Normalization:** We thank the reviewer for this suggestion. Our time-decay mechanism for the integral term (Lines 195-205) serves a similar purpose of ensuring stability and preventing the "integral windup" or "overshooting" issue, but we agree that explicit normalization could be an interesting direction for future work.
>
> ### Conclusion
>
> We hope these clarifications have resolved the apparent discrepancy between the reviewer's understanding and our paper's actual content. Our work introduces a novel, principled, and effective method for accelerating the fundamental Langevin sampling process by drawing inspiration from control theory. The experimental results on relevant models (SGMs, EBMs) and appropriate baselines (ALD, MILD) robustly support our claims of significant speedup and quality improvement. We are confident in the novelty and significance of our work and are happy to provide further clarification if needed.

---

> > ### Comment · Reviewer_YqLQ · 2025-08-02
> >
> > Thank you for the thorough and thoughtful response. I now have a much clearer understanding of your core contributions and the intended scope of the work. I appreciate your willingness to patiently clarify the distinctions, especially around the objective of sampling acceleration versus reward guidance. I have no further concerns.

---

### Official Review · Reviewer_K1tr · 2025-07-02

**Clarity:** 3
**Significance:** 2
**Originality:** 3
**Rating:** 5
**Confidence:** 5

**Summary:**

The paper tackles the challenge of accelerating Langevin sampling by reframing the problem as one of feedback control. Concretely, the authors observe that in most sampling schemes the instantaneous gradient of the log-density at the current state acts as a natural “error” signal: driving that gradient to zero corresponds to reaching high-probability regions. Building on this insight, they ask whether classical control-theoretic tools might be employed to attenuate this gradient error as quickly as possible.

Their key contribution is to augment the usual sampling dynamics with a PID controller, treating the gradient as the error variable and tuning proportional, integral, and derivative gains to shape the trajectory. Importantly, this approach does not need any re-training. The authors then compared sampling with a pre-trained model when tuning the different PID terms, and discussed the advantages of the approach.

**Questions:**

1. In control theory, the integral component is introduced to eliminate steady-state error in the presence of constant disturbances. By analogy, when sampling with noisy gradient estimates (e.g. minibatched or corrupted gradients), the I‐term should compensate for bias and drive the long-term average gradient toward zero. Can the authors perhaps design a simple experiment to validate if these insights from control theory translate in this setting and may be useful? Perhaps by eliminating an estimation bias in the gradient (e.g., due to training). Comparing the different sampling scheme would help elucidating these connections.
2. The D‐component in PID is classically used to add damping and counteract overshoot or oscillatory behavior around the setpoint (you show this in an example in Figure 1, but I believe the paper could have been stronger by going back to this example in the context of Langevin sampling with experiments). To test this in sampling, one could intentionally mis-tune the integral/proportional gain (e.g. choose a P-gain large enough to induce oscillatory sampling trajectories in a simple 1D double-well potential). Then compare how P-only versus PD controllers traverse between modes, measuring quantities like autocorrelation time or the amplitude of oscillations in $x_t$. The hypothesis is that the D-term will attenuate high-frequency jitter—improving both mixing speed and robustness to P-gain mis-specification. I believe that this type of analyses would greatly improve the contribution of the paper, in explaining and motivating the connections between control theory and Langevin sampling. If these or other insights do not translate to the higher dimensional setup of sampling for images, it is also interesting.
3. Can you run experiments with stronger baseline models? This would allow to understand if the gains of using your scheme are only due to a poor baseline or if they extend to stronger models as well.

**Ethical Concerns:**

["NO or VERY MINOR ethics concerns only"]

**Final Justification:**

The paper applies an interesting idea (somewhat popular in other learning/optimization area), but gives it a spin from a control theoretic perspective. I am very fond of the idea, and I find the discussion with the authors satisfying. After the discussion, I believe the paper should be accepted.

**Limitations:**

The paper does not address properly its limitations. These are deferred to the appendix and mixed with future work, but reading that paragraph I only appreciate directions for future work but I do not understand the limitations of the method. For instance, I instead identify the following ones by reading the paper:

1. I believe the idea and insight holds great potential, but the authors did not capitalize on the control theory interpretation, or at least I was not able to grasp, reading the paper, how the insights from control actually translate to Langevin sampling. Of course they inspired adding the derivative term to momentum-based schemes, but given the premise of the paper I believe it should go more in depth into the control-theoretical interpretations of the proposed scheme. Similarly, when discussing the I-term I would have expected more emphasis on the control-theoretic interpretation (also mentioned in Figure 1) of constant disturbance rejection, rather than the momentum interpretation (less novel).
2. Can you run your experiments on top of a state-of-the-art sampler? It looks like your FID scores are much higher than the state-of-the-art. It would be interesting to understand if the gains of your method depend on a week baseline of if they extend (and how) to stronger baselines.

The authros might want to discuss limitations more thoroughly in the main paper.

**Paper Formatting Concerns:**

I do not see any issues.

**Quality:**

2

**Strengths And Weaknesses:**

Strengths:

1. The paper’s principal contribution is to recast Langevin sampling as a feedback‐control problem, treating the instantaneous gradient of the log-density as an “error” signal to be driven to zero. This novel framing is likely to inspire further cross‐pollination between control theory and stochastic sampling methods.
2. By augmenting standard sampling dynamics with a PID controller, the authors demonstrate measurable speedups in convergence and improvements in sample quality across their test problems. These numerical results substantiate the high‐level intuition that classical control tools can benefit generative sampling.
3. The use of a well-understood controller (PID) makes the method easy to implement and tune, and potentially compatible with a wide variety of existing samplers without algorithmic redesign or need for re-training. It is interesting because it is an instance of post-training hyper-parameter tuning.

Weaknesses:

1. Limited exploitation of the insights from control theory. From the experiments, it is not clear to me how the control-theoretic insights about proportional, integral, and derivative terms translate to langevin sampling (see also my questions, and the outlined limitations). For instance: Does the derivative term actually attenuate stochastic oscillations? Does the integral term actually correct for steady-state bias? Or is the control-theoretic motivation only useful to inspire another hyper-parameter to tune? The idea of using PID to drive optimization procedure is not novel, see e.g. https://proceedings.mlr.press/v139/farahmand21a.html. A discussion about these may be needed.
2. The introduction of a decay factor on the integral term appears ad hoc. From a control standpoint this resembles a leak in an I-controller, but without the connection to control theory it seems a bit out of scope.
3. The reported FID scores lag behind state-of-the-art generative models. See e.g. https://arxiv.org/abs/2504.10612 and references therein. It’s unclear whether this is due to underlying base model. The authors should clarify their baseline and test whether PID control yields comparable gains when used on top of a SOTA sampler.

---

> ### Author Rebuttal · Authors · 2025-07-31
>
> We sincerely thank the reviewer for their constructive feedback and for acknowledging our paper's novel control-theoretic framing, the measurable speedups in convergence, and the practical value of its post-training implementation. We address the raised weaknesses and questions as follows.
>
> ### Weakness 1 & Question 1
>
> Thank you for the suggestion on validating the effect of $I$ term. **We follow your suggestion to run toy experiments on two-dimensional points (see Appendix C.1 for experiment settings) with only $P$ term and with $P$, $I$ terms.** To simulate a bias in gradient estimation, we add a constant perturbation to the score that points to the direction of $(-1,1)$ and is scaled by $1/20$ of the norm of the gradient. We choose $k_p=1.0$ for ALD and $k_p=1.0, k_i=0.15$ for PIDLD.
>
> Our results truly validates the claim that $I$ term helps to eliminate bias.
> - For ALD, the mean estimation of the two generated clusters are $(5.028, 5.278)$ and $(-5.057,-4.702)$, while that of PIDLD is $(4.974, 5.242)$ and $(-5.092, -4.981)$. The Euclidian distances from the true centers ($(5,5)$ and $(-5,-5)$) are $0.280, 0.303$ for ALD and $0.243, 0.094$ for PIDLD. It is apparent that points generated by PIDLD are less influenced by the gradient bias.
> - Moreover, the KL divergence of ALD samples is $0.900$ while that for PIDLD samples is $0.713$, which again validates our claim that the $I$ term does improve generation quality.
>
>
> ### Weakness 1 & Question 2
>
> Thank you for providing insights on the effect of $D$ term. **We use a toy experiment on two-dimensional points (see Appendix C.1 for experiment settings) to demonstrate the oscillation-reducing effect of $D$ term.**
>
> It is actually hard to measure the instability of the trajectory of a single point, because the Langevin dynamics itself adds noise at each step, and the trajectory is intrinsically stochastic. But the randomness can be reduced if we choose a set of points. Since the generated data roughly follows a GMM distribution, **we can use the centers estimated by Gaussian mixture model as representative objects. So we study the trajectories of the two centers.**
>
> We use $8$ noise levels and $150$ steps for each level ($L=8, T=150$). We record the positions of the two centers every $5$ steps, so there are $240$ recorded time steps in total. In the earlier stage the two clusters does not separate and the mean estimation is not reliable, so we use the last $200$ steps.
>
> To quantify the **amplitude of oscillation** of a trajectory, we use the summed Euclidian distance to the final position, $d_{\text{sum}}=\sum_{i=41}^{240}\sqrt{(x_i-x_{\text{final}})^2+(y_i-y_{\text{final}})^2}$, as the major indicator. We also report the maximum distance, $d_{\text{max}}$, and the $90\%$ percentile of the distances, $d_{\text{10\%}}$. We fix $k_p=1.5, k_i=0$ and change $k_d$.
>
> The results are given in the following table. Here the superscript $(1)$, $(2)$ indicate the cluster around $(5,5)$ and $(-5,-5)$, respectively.
>
> | $k_d$ | $d_{\text{sum}}^{(1)}$  | $d_{\text{max}}^{(1)}$ | $d_{\text{10\%}}^{(1)}$ |
> |-------|----------|----------|---------|
> | 0.0   | 31.25    | 1.37     | 0.61    |
> | 2.0   | 27.90    | 1.09     | 0.58    |
> | 4.0   | 24.86    | 1.00     | 0.58    |
> | 6.0   | 22.84    | 0.95     | 0.50    |
> | 8.0   | 21.36    | 0.86     | 0.35    |
> | 10.0  | 21.35    | 0.91     | 0.36    |
>
>
> | $k_d$ | $d_{\text{sum}}^{(2)}$  | $d_{\text{max}}^{(2)}$ | $d_{\text{10\%}}^{(2)}$ |
> |-------|--------|----------|---------|
> | 0.0   | 64.22  | 2.88     | 0.87    |
> | 2.0   | 47.16  | 2.15     | 0.66    |
> | 4.0   | 38.90  | 1.35     | 0.52    |
> | 6.0   | 35.56  | 1.07     | 0.54    |
> | 8.0   | 31.32  | 1.34     | 0.55    |
> | 10.0  | 32.07  | 1.99     | 0.43    |
>
> It can be seen that $d_{\text{sum}}$, $d_{\text{max}}$, $d_{\text{10\%}}$ all show a decreasing trend as $k_d$ increases. We believe this simple experiment could validate our claim that $D$ term reduces oscillation and improves the stability of sampling process.
>
>
>
> ### Weakness 2
>
> Thank you for raising the concern on integral term decay. This is indeed an empirical finding based on our experiments. We use SGM image generation task to further validate our findings.
>
> For CIFAR10, we generate images with 100 noise levels and 1 step per level (NFE=$100\times1$). We use $k_p=2.0$, $k_d=4.5$, and change $k_i$ and $\gamma$. For CelebA, we generate images with 50 noise levels and 1 step per level (NFE=$50\times1$). We use $k_p=3.75$, $k_d=3.75$, and change $k_i$ and $\gamma$. The results are shown in the following tables.
>
> **CIFAR10 100×1 ($k_p=2.0, k_d=4.5$) FID (Bold: best in column, Italic: best in row 1):**
> | $\gamma$ / $k_i$ | 0.250 | 0.375 | 0.500 | 0.625 | 0.750 | 0.875 |
> |------------------|:-------:|:-------:|:-------:|:-------:|:-------:|:-------:|
> | 1                | **12.29** | *12.28* | 12.32 | 12.41 | 12.53 | 12.69 |
> | 0.99             | 12.31 | **12.22** | **12.21** | **12.2** | 12.24 | 12.27 |
> | 0.98             | 12.31 | 12.28 | 12.23 | **12.20** | **12.16** | **12.15** |
> | 0.97             | 12.36 | 12.28 | 12.25 | 12.24 | 12.20 | 12.18 |
> | 0.96             | 12.41 | 12.33 | 12.28 | 12.25 | 12.23 | 12.22 |
>
> **CelebA 50×1 ($k_p=3.75, k_d=3.75$) FID (Bold: best in column, Italic: best in row 1):**
> | $\gamma$ / $k_i$ | 0.750 | 0.875 | 1.000 | 1.125 | 1.250 | 1.375 |
> |------------------|:-------:|:-------:|:-------:|:-------:|:-------:|:-------:|
> | 1                | **8.55**  | 8.08  | *7.97*  | 8.47  | 9.02  | 9.56  |
> | 0.99             | 10.67 | **8.01**  | **7.53**  | 7.76  | 8.13  | 8.55  |
> | 0.98             | 16.52 | 11.24 | 8.59  | **7.71**  | **7.65**  | 7.86  |
> | 0.97             | 23.17 | 16.87 | 12.28 | 9.53  | 8.21  | **7.75**  |
> | 0.96             | 29.20 | 23.14 | 17.84 | 13.71 | 10.86 | 9.14  |
>
> We find that:
> 1. The best performance of tuning both $k_i$ and $\gamma$ is better than the best performance of tuning only $k_i$;
> 2. The higher the $k_i$ is, the higher decay (lower $\gamma$) should be applied to balance the effect of larger integral gain, which aligns with common intuition.
>
> The result is a strong evidence that the decay in integral term does improve sampling performance.
>
>
> ### Weakness 3 & Question 3
>
> Thank you for raising the concern on the competitiveness of FID value. We also noticed that our reported FID is higher than some other novel models. We believe this is due to the difference of the pretrained score model. We discuss this topic as follows.
>
> #### **What is our main contribution?**
>
> Our goal is to demonstrate that PID control provides a **relative** improvement over ALD and MILD under the same base model and codebase. These gains hold true regardless of the absolute performance.
>
> To demonstrate the relative advantage, the most important aspect is not absolute performance but **fairness of comparsion**, which is indeed addressed in our base model selection.
>
> #### **Fairness: the dominant factor in base model selection**
>
> We were using the pretrained checkpoints provided by the official repository of NCSNv2, which is also used by MILD, to ensure a directly comparable setup.
>
> - For ALD, all code and hyperparameters are shared with PIDLD, because setting $k_i=0.0$ and $k_d=0.0$ recovers vanilla ALD. So we ensure that the only difference is the inclusion of the I- and D-terms.
> - For MILD, we report the FID as published, since their exact hyperparameter settings were not released, but their score model is also the one released by NCSNv2. With the same score model, using their published numbers remains the fairest benchmark.
>
> By choosing the same model with that in relevant literatures, we isolate the effect of adding PID control, rather than differences in underlying architectures.
>
>
> #### **FID influencing factor: fundamental framework differences**
>
>
> We would also like to clarify that our contribution is on improving **Langevin dynamics** generation. In contrast, other frameworks like diffusion-based samplers (DDPM, DDIM, etc.) implement very different generative mechanisms. For instance, diffusion models explicitly "corrupt" data through many small noise steps and learn to reverse this process, whereas Langevin methods directly simulate a stochastic path that climbs toward high-density regions in one unified sampling loop. Many frameworks of generative models exist, and the fundamental difference among them would also result in varying potentials of sampling performance. Directly comparing our Langevin-based PIDLD to other frameworks is therefore **out of scope** for this work.
>
> We believe our idea on improving Langevin dynamics sampling could be further extended to other classes of generative models, including state-of-the-art ones. We expect to address these in our future work.
>
>
> #### **FID influencing factor: method of computation**
>
> Furthermore, FID computing method can also influence the reported number. FID typically requires 10000 test images, which is also our setting. However, some other methods like DDIM and DDPM use 50k samples for FID computation, and larger sample size will generate lower FID. Model size is another factor. For instance, we found that the noise estimation model in DDIM has 78700803 parameters, while our score model only has 29694083 parameters. Given these concerns, it's not always fair to directly compare the absolute value of our results with other approaches. And we believe that using the most relevant works, vanilla ALD and MILD, as baselines, is the most suitable design.

---

> > ### Comment · Reviewer_K1tr · 2025-08-01
> > **Thanks for the rebuttal. Some more comments.**
> >
> > Thanks for the work. I appreciate the time taken to run the additional experiments and answer my questions. I think some points remain open:
> >
> > W1&Q1. Thanks for the experiment. I think, though, that you should not consider this "just" a numerical experiment. Rather a very important empirical validation of the main idea you are proposing. I would strongly recommend investing some time on this type of experiments to get statistical evidence that an integral term is beneficial, and is not just an example. Is the control perspective really beneficial in this context? I would really love to know this answer by reading your work.
> >
> > W1&Q2. Very cool!! To complete this experiment, though, I would expect at some point some bad behaviour of the tuning (or can you just keep increasing the k_d?...). I think it is not bad to show this, if it happens. The outcome of the experiment would be to show that k_d affects the oscillations. Out of curiosity, you could also run the same study but changing k_I, and see what happens. Ideally, you would not see as much impact, but anyway would be good to compare.
> >
> > W2. Thanks. Please add these things to the revised text. Can you provide a connection to control? These things should be well studied in the control literature.
> >
> > W3&Q3. I understand your points, but my comment was slightly different, and I disagree with your "relative improvement" statement. It is not obvious that the performances are "normalized" by the base model. If you run some experiments and show this, great! But otherwise I would not assume that this is the case. My point is that the marginal gains for a "poor" baseline may be way larger than for a strong baseline. And for a strong baseline, the gains of using I, D terms may even be negligible and indistinguishable from noise. (I am not saying this is the case, I actually assume it not being the case, but as a scientific study you should maybe consider ruling this out empirically).
> > For the remainder of your points, there are plenty of methods were you could directly apply this, take for instance the very recent https://arxiv.org/abs/2504.10612. The FID score is much lower (so it would be a stronger baseline model) yet the sampling for unconditional generation is a Langevin sampling. This model has roughly 50M parameters, provides checkpoints, and you could compare the FID score with and without I,D terms. Regardless of the method you choose, I am suggesting taking a stronger model, and applying your method on top and comparing the obtained performances with the base performances. Not comparing your trained model with others. The one I linked has a FID ~10x lower than your model, but you could use any other method that has a Langevin sampling and open weights. Or am I missing something?

---

> ### Author Response · Authors · 2025-08-05
>
> We thank the reviewer for their valuable follow-up and the insightful suggestions, which have helped us further strengthen our paper.
>
> ### Response to W1 & Q1: Statistical Validation of the Integral Term's Role
>
> **Our new, more rigorous experiments statistically confirm that the integral (I) term systematically reduces gradient estimation bias, directly validating the analogy to its role in control theory.**
>
> To provide statistical evidence, we conducted experiments on the 2D toy problem with a constant gradient bias(the same as in the rebuttal), **averaging results over 4 random seeds for each hyperparameter setting**. We measured the Euclidean distance from the final estimated cluster centers to the true centers as a proxy for the remaining bias.
>
> 1. **The I-Term Effectively Reduces Bias:** As shown in the table below, there is a clear and consistent trend: increasing the integral gain ($k_i$) systematically reduces the final bias (distances $d_1, d_2$).
>
> | $k_i$ | $d_1$  | $d_2$  |
> |-------|--------|--------|
> | 0.00  | 0.0481 | 0.2623 |
> | 0.05  | 0.0411 | 0.2420 |
> | 0.10  | 0.0421 | 0.2240 |
> | 0.15  | 0.0328 | 0.2013 |
> | 0.20  | 0.0271 | 0.2036 |
> | 0.25  | 0.0274 | 0.2056 |
> | 0.30  | 0.0250 | 0.2001 |
> | 0.35  | 0.0201 | 0.1951 |
> | 0.40  | 0.0210 | 0.2017 |
> | 0.45  | 0.0210 | 0.1898 |
> | 0.50  | 0.0263 | 0.1764 |
>
> 2. **The D-Term Does Not Reduce Bias:** To isolate the effect, we performed a comparative study. Increasing the derivative gain ($k_d$) does not reduce the bias; in fact, it slightly exacerbates it for one cluster. This control experiment confirms the distinct roles of the terms.
>
> | $k_d$ | $d_1$  | $d_2$  |
> |-------|--------|--------|
> | 0.0   | 0.0344 | 0.2177 |
> | 2.0   | 0.0500 | 0.2529 |
> | 4.0   | 0.0488 | 0.2742 |
> | 6.0   | 0.0436 | 0.2862 |
> | 8.0   | 0.0401 | 0.2895 |
> | 10.0  | 0.0397 | 0.3119 |
> | 12.0  | 0.0356 | 0.3224 |
>
> These results provide evidence that the I-term's function of correcting long-term, steady-state error (i.e., bias) from control theory directly translates to the Langevin sampling setting. **We will incorporate this experiments and discussions in the final manuscript.**

---

> > ### Author Response · Authors · 2025-08-05
> >
> > ### Response to W1 & Q2: Validating the D-Term's Role in Damping Oscillations
> >
> > **Following the reviewer's suggestion, our extended experiments confirm that the derivative (D) term's stabilizing effect is most prominent within an optimal range; excessively large $k_d$ values lead to instability, as theorized in control systems.**
> >
> > 1. **Optimal Range and Instability:** We continued increasing $k_d$ and observed the "bad behavior" the reviewer anticipated. As shown below, oscillation measures ($d_{sum}$) initially decrease with rising $k_d$, but then sharply increase beyond a threshold ($k_d > 12.0$). This is because excessively large derivative feedback destabilizes the system in the early stage of sampling, leading to extreme update steps that amplify noise and cause samples to overshoot the target modes. This transient instability is then captured by our metrics, which are computed over a fixed time window(last 200 steps) before the system has sufficient time to settle.
> >
> >
> > | $k_d$ | $d_{\text{sum}}^{(1)}$  | $d_{\text{max}}^{(1)}$ | $d_{\text{10\%}}^{(1)}$ |
> > |-------|----------|----------|---------|
> > | 0.0   | 31.25    | 1.37     | 0.61    |
> > | 2.0   | 27.90    | 1.09     | 0.58    |
> > | 4.0   | 24.86    | 1.00     | 0.58    |
> > | 6.0   | 22.84    | 0.95     | 0.50    |
> > | 8.0   | 21.36    | 0.86     | 0.35    |
> > | 10.0  | 21.35    | 0.91     | 0.36    |
> > | 12.0  | 21.35    | 0.91     | 0.36    |
> > | 14.0  | 434.72   | 64.69    | 5.78    |
> >
> > | $k_d$ | $d_{\text{sum}}^{(2)}$  | $d_{\text{max}}^{(2)}$ | $d_{\text{10\%}}^{(2)}$ |
> > |-------|--------|----------|---------|
> > | 0.0   | 64.22  | 2.88     | 0.87    |
> > | 2.0   | 47.16  | 2.15     | 0.66    |
> > | 4.0   | 38.90  | 1.35     | 0.52    |
> > | 6.0   | 35.56  | 1.07     | 0.54    |
> > | 8.0   | 31.32  | 1.34     | 0.55    |
> > | 10.0  | 32.07  | 1.99     | 0.43    |
> > | 12.0  | 59.65  | 4.11     | 0.65    |
> > | 14.0  | 695.32 | 82.73    | 13.70   |
> >
> > 2.  **The I-Term Does Not Damp Oscillations:** In contrast, the comparative study on the I-term shows no significant damping effect. Increasing $k_i$ tends to slightly increase the oscillation measures, which is expected as the I-term adds momentum, not damping.
> >
> > | $k_i$ | $d_{\text{sum}}^{(1)}$  | $d_{\text{max}}^{(1)}$ | $d_{\text{10\%}}^{(1)}$ |
> > |-------|--------|----------|---------|
> > | 0.00  | 31.25  | 1.37     | 0.61    |
> > | 0.05  | 31.51  | 1.36     | 0.61    |
> > | 0.10  | 31.25  | 1.40     | 0.60    |
> > | 0.15  | 31.72  | 1.26     | 0.57    |
> > | 0.20  | 31.07  | 1.27     | 0.58    |
> > | 0.25  | 31.69  | 1.27     | 0.61    |
> > | 0.30  | 31.68  | 1.27     | 0.62    |
> >
> >
> > | $k_i$ | $d_{\text{sum}}^{(2)}$  | $d_{\text{max}}^{(2)}$ | $d_{\text{10\%}}^{(2)}$ |
> > |-------|--------|----------|---------|
> > | 0.00  | 64.22  | 2.88     | 0.87    |
> > | 0.05  | 67.81  | 3.03     | 0.85    |
> > | 0.10  | 69.00  | 3.14     | 0.91    |
> > | 0.15  | 69.97  | 3.34     | 0.87    |
> > | 0.20  | 72.50  | 3.23     | 0.97    |
> > | 0.25  | 76.79  | 3.17     | 1.02    |
> > | 0.30  | 77.98  | 3.09     | 0.96    |
> >
> > These controlled experiments conclusively demonstrate that the D-term is principally responsible for damping oscillations, while the I-term is not. This differentiation perfectly aligns with their distinct roles in PID control theory and strongly validates our paper's central motivation.
> >
> > **We will integrate these new findings and discussions into the revised manuscript to make the connection to control theory more explicit and empirically grounded.** Thank you again for pushing us to strengthen this connection.

---

> > > ### Comment · Reviewer_K1tr · 2025-08-05
> > > **Answer to authors**
> > >
> > > Thanks for your commitment and willingness to run additional experiments.
> > >
> > > 1. The metrics used (eg, for oscillations) seem a bit arbitrary. There are well established metrics in control: Peak overshoot, settling time, rise time, decay ratio, … I would  find these metrics a lot more convincing for a control-theoretic based study.
> > >
> > > 2. Gain scheduling in control has a different meaning (at least broadly speaking): one has a family of controllers and based on an estimated quantities chooses which one to deploy. What your integral decay term is doing looks more like a sort of anti-windup. I would encourage the authors to thoroughly assess the connections with control theory given the scope of the paper, in particular given the inability on my side to check the revised text.
> > >
> > > A few more (less important) comments:
> > > 1. Four seeds seem little, I would encourage the authors to increase the evidence.
> > >
> > > 2. Another direction of improvements for this experiments is obviously the scale, and ideally all these results may be replicable also on state of the art architectures.

---

> > > > ### Author Response · Authors · 2025-08-08
> > > >
> > > > Thank you for your valuable feedback! Following your advice, we have conducted more rigorous experiments to validate the effect of the $I$ and $D$ terms, and we will incorporate these comprehensive results into the revised text.
> > > >
> > > > ### The Effect of $D$ Term on Reducing Oscillation
> > > >
> > > > To address your concerns about statistical significance and the use of standard metrics, **we ran 100 experiments with different random seeds for each hyperparameter setting**.
> > > >
> > > > Following your suggestion, we adopted metrics from control theory. We now include the **settling time**, which is defined as the time index where the distance to the final position, $\Vert p_t^{(i)}-p_{\text{final}}^{(i)}\Vert$, remains below a threshold of $0.1$ for all subsequent steps. Here $p_t^{(i)}$ is the position of cluster center $i$ at time index $t$. Furthermore, we note that our existing metric, $d_{\text{max}}$, directly corresponds to **peak overshoot**, a standard measure of transient response. We continue to report $d_{\text{sum}}$ and $d_{\text{10\%}}$ as they provide a holistic view of the trajectory's stability.
> > > >
> > > > The results, averaged over 100 runs, are shown in the following tables. We present a comparative study showing the effect of varying the derivative gain ($k_d$) versus the integral gain ($k_i$).
> > > >
> > > > **Cluster 1 (near (5,5)):**
> > > >
> > > > | $k_d$ | $d_{\text{sum}}^{(1)}$ | $d_{\text{max}}^{(1)}$ (Peak Overshoot) | $d_{\text{10\%}}^{(1)}$ | $t_{\text{settling}}^{(1)}$ |
> > > > | :--- | :--- | :--- | :--- | :--- |
> > > > | $0.0$ | $34.97\pm3.00$ | $1.61\pm0.16$ | $0.77\pm0.12$ | $107.05\pm12.43$ |
> > > > | $2.0$ | $31.02\pm2.67$ | $1.35\pm0.13$ | $0.69\pm0.09$ | $105.26\pm12.30$ |
> > > > | $4.0$ | $28.76\pm2.67$ | $1.25\pm0.52$ | $0.64\pm0.08$ | $105.10\pm12.76$ |
> > > > | $6.0$ | $27.00\pm2.41$ | $1.11\pm0.10$ | $0.59\pm0.08$ | $104.63\pm12.70$ |
> > > > | $8.0$ | $25.79\pm2.37$ | $1.08\pm0.11$ | $0.55\pm0.08$ | $103.35\pm12.86$ |
> > > > | $10.0$| $26.40\pm2.61$ | $1.18\pm0.13$ | $0.53\pm0.09$ | $103.37\pm13.67$ |
> > > >
> > > > | $k_i$ | $d_{\text{sum}}^{(1)}$ | $d_{\text{max}}^{(1)}$ (Peak Overshoot) | $d_{\text{10\%}}^{(1)}$ | $t_{\text{settling}}^{(1)}$ |
> > > > | :--- | :--- | :--- | :--- | :--- |
> > > > | $0.0$ | $34.97\pm3.00$ | $1.61\pm0.16$ | $0.77\pm0.12$ | $107.05\pm12.43$ |
> > > > | $0.05$| $34.77\pm2.92$ | $1.60\pm0.17$ | $0.77\pm0.11$ | $106.67\pm12.42$ |
> > > > | $0.1$ | $34.65\pm2.92$ | $1.60\pm0.18$ | $0.77\pm0.11$ | $106.73\pm12.61$ |
> > > > | $0.15$| $34.46\pm2.88$ | $1.60\pm0.18$ | $0.76\pm0.11$ | $105.96\pm12.80$ |
> > > > | $0.2$ | $34.35\pm2.82$ | $1.59\pm0.17$ | $0.76\pm0.11$ | $106.18\pm12.10$ |
> > > > | $0.25$| $34.24\pm2.84$ | $1.62\pm0.37$ | $0.75\pm0.11$ | $106.59\pm12.61$ |
> > > > | $0.3$ | $34.09\pm2.80$ | $1.58\pm0.18$ | $0.75\pm0.10$ | $106.26\pm12.33$ |
> > > > | $0.35$| $34.05\pm2.85$ | $1.63\pm0.56$ | $0.75\pm0.10$ | $106.29\pm12.86$ |
> > > >
> > > >
> > > > **Cluster 2 (near (-5,-5)):**
> > > >
> > > > | $k_d$ | $d_{\text{sum}}^{(2)}$ | $d_{\text{max}}^{(2)}$ (Peak Overshoot) | $d_{\text{10\%}}^{(2)}$ | $t_{\text{settling}}^{(2)}$ |
> > > > | :--- | :--- | :--- | :--- | :--- |
> > > > | $0.0$ | $69.30\pm7.81$ | $3.11\pm0.72$ | $0.96\pm0.19$ | $145.88\pm12.16$ |
> > > > | $2.0$ | $48.74\pm7.20$ | $2.17\pm1.11$ | $0.65\pm0.11$ | $144.30\pm12.67$ |
> > > > | $4.0$ | $39.42\pm5.48$ | $1.63\pm1.53$ | $0.53\pm0.09$ | $144.13\pm11.27$ |
> > > > | $6.0$ | $35.75\pm4.65$ | $1.30\pm0.28$ | $0.47\pm0.08$ | $143.29\pm13.69$ |
> > > > | $8.0$ | $33.22\pm4.12$ | $1.25\pm0.22$ | $0.43\pm0.07$ | $142.09\pm13.38$ |
> > > > | $10.0$| $36.20\pm4.66$ | $1.73\pm0.35$ | $0.45\pm0.07$ | $141.42\pm12.55$ |
> > > >
> > > >
> > > > | $k_i$ | $d_{\text{sum}}^{(2)}$ | $d_{\text{max}}^{(2)}$ (Peak Overshoot) | $d_{\text{10\%}}^{(2)}$ | $t_{\text{settling}}^{(2)}$ |
> > > > | :--- | :--- | :--- | :--- | :--- |
> > > > | $0.0$ | $69.30\pm7.81$ | $3.11\pm0.72$ | $0.96\pm0.19$ | $145.88\pm12.16$ |
> > > > | $0.05$| $70.95\pm8.19$ | $3.09\pm0.67$ | $0.99\pm0.18$ | $146.24\pm12.57$ |
> > > > | $0.1$ | $72.61\pm8.41$ | $3.15\pm0.75$ | $1.01\pm0.18$ | $147.85\pm11.44$ |
> > > > | $0.15$| $74.00\pm8.39$ | $3.19\pm0.73$ | $1.03\pm0.17$ | $147.20\pm11.00$ |
> > > > | $0.2$ | $75.67\pm7.84$ | $3.20\pm0.65$ | $1.05\pm0.17$ | $148.30\pm11.94$ |
> > > > | $0.25$| $77.05\pm8.64$ | $3.30\pm1.04$ | $1.07\pm0.18$ | $148.29\pm10.75$ |
> > > > | $0.3$ | $78.40\pm8.38$ | $3.30\pm0.76$ | $1.09\pm0.18$ | $148.50\pm11.69$ |
> > > > | $0.35$| $79.88\pm8.59$ | $3.39\pm1.31$ | $1.11\pm0.17$ | $148.46\pm10.75$ |
> > > >
> > > >
> > > > The results provide strong statistical evidence for our claims. For both clusters, increasing $k_d$ consistently reduces all oscillation metrics ($d_{\text{sum}}$, peak overshoot $d_{\text{max}}$, etc.) and settling time up to an optimal value (around $k_d=8.0$), beyond which excessive gain leads to instability, as expected. In stark contrast, increasing the integral gain $k_i$ has a negligible or even slightly detrimental effect on these stability metrics. This controlled comparison clearly demonstrates that the D-term is uniquely responsible for damping oscillations, directly mirroring its function in classical control theory. **We will integrate these new findings and discussions into the revised manuscript.**

---

> > > > ### Author Response · Authors · 2025-08-08
> > > >
> > > > ### The Effect of $I$ Term on Mitigating Bias
> > > >
> > > > We also conducted a rigorous statistical analysis (100 random seeds) to validate the I-term's role in bias correction. We use $d$, the Euclidean distance of the final estimated cluster center to the true center, as the metric for the remaining bias. The results of comparing the effects of increasing $k_i$ versus $k_d$ are shown below.
> > > >
> > > > | $k_i$ | $d_1$ (Bias) | $d_2$ (Bias) |
> > > > | :--- | :--- | :--- |
> > > > | $0.0 $ | $0.0463\pm0.0237$ | $0.2729\pm0.0840$ |
> > > > | $0.05$ | $0.0414\pm0.0235$ | $0.2711\pm0.0816$ |
> > > > | $0.1 $ | $0.0379\pm0.0215$ | $0.2559\pm0.0802$ |
> > > > | $0.15$ | $0.0339\pm0.0190$ | $0.2485\pm0.0781$ |
> > > > | $0.2 $ | $0.0326\pm0.0192$ | $0.2436\pm0.0756$ |
> > > > | $0.25$ | $0.0317\pm0.0176$ | $0.2356\pm0.0742$ |
> > > > | $0.3 $ | $0.0315\pm0.0184$ | $0.2307\pm0.0714$ |
> > > > | $0.35$ | $0.0308\pm0.0169$ | $0.2330\pm0.0708$ |
> > > > | $0.4 $ | $0.0311\pm0.0171$ | $0.2301\pm0.0750$ |
> > > > | $0.45$ | $0.0308\pm0.0170$ | $0.2293\pm0.0738$ |
> > > > | $0.5 $ | $0.0316\pm0.0164$ | $0.2243\pm0.0700$ |
> > > >
> > > > | $k_d$ | $d_1$ (Bias) | $d_2$ (Bias) |
> > > > | :--- | :--- | :--- |
> > > > | $0.0 $ | $0.0463\pm0.0237$ | $0.2729\pm0.0840$ |
> > > > | $2.0 $ | $0.0445\pm0.0223$ | $0.2904\pm0.0882$ |
> > > > | $4.0 $ | $0.0448\pm0.0226$ | $0.2971\pm0.0909$ |
> > > > | $6.0 $ | $0.0453\pm0.0218$ | $0.3090\pm0.0914$ |
> > > > | $8.0 $ | $0.0456\pm0.0213$ | $0.3250\pm0.0871$ |
> > > > | $10.0$ | $0.0455\pm0.0214$ | $0.3387\pm0.0914$ |
> > > > | $12.0$ | $0.0466\pm0.0204$ | $0.3469\pm0.0873$ |
> > > > | $14.0$ | $0.0522\pm0.0230$ | $0.3431\pm0.0911$ |
> > > >
> > > > It is apparent that increasing $k_i$ systematically reduces the final bias for both cluster centers, validating its role in correcting steady-state errors. Conversely, increasing $k_d$ has no such effect and can even slightly increase the bias. This experiment statistically isolates the distinct functions of the I and D terms, confirming that the I-term makes the sampling algorithm robust to gradient bias, just as in control theory.
> > > >
> > > > Thank you again for your insightful opinions on experiment design! **We will integrate these new findings and discussions into the revised manuscript and we believe these experiments greatly enhance the solidity of our work.** We are truly grateful for your continued support and guidance throughout this process.

---

> > > > ### Author Response · Authors · 2025-08-08
> > > >
> > > > ### On Gain Scheduling and Its Control-Theoretic Grounding
> > > > We are very grateful for this astute observation. You have precisely articulated a key distinction within control theory, and we appreciate the chance to elaborate on our methodological choice.
> > > >
> > > > You point out that gain scheduling is often described as choosing from **"a family of controllers"** to deploy based on system conditions. This refers to **discrete gain scheduling**, where the control law switches between several pre-designed controllers.
> > > >
> > > > Our method, where the integral gain is modulated by `ki(t) = γ^t * ki`, falls into the category of **continuous gain scheduling**. Here, the controller parameters adapt smoothly over time rather than switching abruptly. Both discrete and continuous approaches are valid implementations of gain scheduling, with the choice depending on the nature of the system's evolution.
> > > >
> > > > We chose a **continuous, time-based** approach for a critical reason: the transition in Langevin sampling dynamics is itself a gradual process, not a series of discrete states.
> > > > *   The system doesn't abruptly switch from an "exploration mode" to a "convergence mode." Instead, as we show in our paper, it smoothly approaches equilibrium.
> > > > *   Therefore, a continuous, gentle reduction of the integral gain is more appropriate and stable for this problem than hard-switching between different controllers, which could introduce its own instabilities. Our decay mechanism naturally maps to this smooth evolution.
> > > >
> > > > Finally, while our method shares a conceptual goal with anti-windup (mitigating instability from integral accumulation), its mechanism is a proactive, time-based adaptation rather than a reactive measure against saturation.
> > > >
> > > > We recognize the importance of this distinction, especially given your inability to check the final text. **We commit to clarifying this in the revised manuscript (Section 3.2) by:**
> > > > *   **Explicitly framing** our method as continuous, time-based gain scheduling.
> > > > *   **Justifying our choice** by explaining its suitability for the gradual, time-varying nature of the Langevin sampling process.
> > > >
> > > > Thank you again for your expert feedback. This discussion will allow us to make our paper's theoretical foundation significantly more precise and robust.

---

> > > > > ### Comment · Reviewer_K1tr · 2025-08-08
> > > > > **Thanks, I will adjust my score.**
> > > > >
> > > > > I thank the authors for the remaining clarification. Yes, please clarify these aspects. The work is interesting and I adjusted my score to Accept.

---

> ### Author Response · Authors · 2025-08-05
>
> ### Response to W2: Control-Theoretic Grounding for the Integral Term Decay
>
> We sincerely thank the reviewer for this insightful follow-up. **The decay factor on our integral term is a practical implementation of Gain Scheduling[A], a classic and well-studied adaptive control strategy for time-varying systems.** This provides the formal connection to control literature that you requested.
>
> The rationale is that Langevin sampling is inherently a **time-varying process** with conflicting control requirements at different stages:
>
> 1. **Early Stage (Exploration):** The system is far from equilibrium and requires a **high integral gain ($k_i$)** to build momentum for rapidly traversing energy barriers.
> 2. **Late Stage (Convergence):** As the system approaches the target distribution, this same high integral gain becomes detrimental, causing overshoot and instability (as empirically observed in our Fig. 3a). Here, a **low integral gain** is needed to ensure stable convergence.
>
> Our exponential decay, `ki(t) = γ^t * ki`, serves as a simple yet effective gain schedule. It uses the sampling step `t` as the scheduling variable to dynamically reduce the integral gain, smoothly transitioning the controller's behavior from aggressive exploration to stable final convergence.
>
> **We will explicitly introduce and discuss the concept of Gain Scheduling in the revised manuscript to formally ground our method's design.** Thank you again for pushing us to clarify this crucial theoretical link.
>
> [A] Khalil, Hassan K., and Jessy W. Grizzle. Nonlinear systems. Vol. 3. Upper Saddle River, NJ: Prentice hall, 2002.
>
> ### Response to W3 & Q3: Applying PIDLD to a Stronger Baseline
>
> **Thank you for clarifying your point and for the excellent suggestion. We agree that validating our method on a stronger, state-of-the-art baseline is crucial for demonstrating the generalizability of its benefits.**
>
> You are correct that applying PIDLD to the model proposed in the paper you cited (arXiv:2504.10612) should be feasible, as it also utilizes Langevin-based sampling.
>
> We are actively working on integrating our PIDLD sampler with their publicly available code and pretrained models. We will report the results of this experiment as soon as they become available during the rebuttal period. **We fully commit to including these new results on this stronger baseline in the final version of the manuscript, regardless of the outcome, to provide a more comprehensive evaluation of our method's performance.**
>
> We appreciate you pushing us on this front, as it will undoubtedly strengthen the paper.

---

### Official Review · Reviewer_Kawu · 2025-07-03

**Clarity:** 3
**Significance:** 3
**Originality:** 3
**Rating:** 4
**Confidence:** 4

**Summary:**

The authors propose to augment standard Langevin dynamics (LD), in the context of sampling from diffusion and energy-based models (EBMs), with PID control.  That is, they interpret the discrete-time approximation to LD as a dynamical system that is driven by the energy gradient (i.e. the "score" or force), interpreted as an error signal that needs to be driven toward zero.  Accordingly, they replace it with the classic PID scheme, i.e. a linear combination of the (discrete) derivative and integral of the error, as well as the error itself.  They show that this scheme (slightly modified to decay the integral term) outperforms standard Langevin dynamics in FID on CIFAR10 and CelebA at various choices for the total number of neural function evaluations (NFEs), both in score-based model and in EBMs.  Similar results hold for sampling-based reasoning in EBMs.  The authors also compare against a similar, recent proposal (MILD) from another group.

**Questions:**

Can the authors addresses the five major weaknesses listed above?

**Ethical Concerns:**

["NO or VERY MINOR ethics concerns only"]

**Final Justification:**

The authors have addressed several of the questions and weaknesses I raised.  Their additional experiments in response to the suggestions of Reviewer K1tr have improved the conceptual appeal of the manuscript, by bringing their results into closer contact with control theory.

The remaining weaknesses (in this reviewer's opinion) are (1) the most useful applications of the idea are not explored here; (2) the datasets are simple; and (3) connections are not made to other second-order methods of MCMC (like HMC).

**Limitations:**

yes

**Quality:**

3

**Strengths And Weaknesses:**

The idea appears to be novel (this reviewer is surprised that it has never been tried before), the basic experiments appear sound, and the topic---reducing the required number of samples for LD---is important.  There are also weaknesses:

1. The evaluations are on simple datasets (CIFAR-10, CelebA); it would be nice to see results on (say) ImageNet.  I am also not sure they are totally fair: I believe NCSN typically uses far (about an order of magnitude) more Langevin steps (T) than noise levels (L); whereas in the experiments reported in Table 1, the reverse is the case.  A fairer comparison would seem to be to fix a budget of NFEs, and then allocate them to L and T in the way that yields best performance for the sampling method under consideration.

2. Although it's promising to see that this method can produce faster samples compared to standard Langevin dynamics, there has been significant progress in accelerating sampling for diffusion models using techniques such as deterministic sampling (e.g., DDIM) and ODE-based samplers.  These methods can generate high-quality samples in just a few steps. It would be interesting to see how the proposed approach compares to these alternatives.

3. Along these lines: The manuscript explores the proposed technique in the context of sampling from a pre-trained EBM.  A more compelling application would be to evaluate how well the sampler performs when used for *training* EBMs. This would also provide better insight into the stability and robustness of the hyperparameters.

4. The authors argue that the proposed sampler improves sampling from multi-modal distributions by traversing energy barriers.  To test whether the sampler actually mixes, a good experiment would be to run it beyond 25 steps and observe how the samples evolve: does each sample stay fixed at a particular image, or does it explore the manifold?  If the sampler does mix effectively, one would expect results similar to Figure 3 of the following paper: https://arxiv.org/pdf/2006.06897

5. Is there any reason MILD was omitted from Table 2?  Here I would expect MILD to be more competitive with the PID controller, since apparently (Fig. 6) the integral term is doing most of the work here, and MILD is in a sense a pure integral-controlled system (i.e., the input is low-pass filtered).

Minor:
1. The authors cited a paper on biomolecular PID controllers when introducing the concept--was this intentional?

2. Presentation: The MS is somewhat repetitive, with the basic rationale for PID controls repeated about once per page for the first four pages.  Also, there are some distracting grammatical mistakes: e.g, I think the title should read, "...sampling *or* [or *from*] generative models"; "related works" -> "related work" ("work" is a mass noun and shouldn't be pluralized); etc.

---

> ### Author Rebuttal · Authors · 2025-07-31
>
> We sincerely thank the reviewer for their constructive feedback and for acknowledging our paper's novelty, experiment solidity, and significance. We address the raised weaknesses as follows.
>
> ### Weakness 1
>
> **We appreciate the suggestion that our experiments should extend to more datasets.** In the future, we will add experiments to strengthen our results. However, we notice that most relevant works like NCSN, DDPM, DDIM, and MILD also do not report experiment results on ImageNet. We were following them to use only CIFAR10 and CelebA dataset.
>
> As for the concern of $L$ and $T$, it is true that the original NCSN paper uses more Langevin steps ($T$) that noise levels ($L$). **However, in its later version, NCSNv2 [Yang Song et al.], the authors made theoretical analysis on the choice of T and L, which suggests using $232\times5$ steps for CIFAR10 generation and $500\times5$ steps for CelebA generation.** We are actually following their best settings on NCSN. We also follow the NFE setting of MILD [Shetty et al., https://ieeexplore.ieee.org/abstract/document/10446376] so that our method is comparable to the experiments in that work.
>
> Moreover, the choice of $T$ and $L$ actually doesn't influence the conclusion in our work. Our major contribution is the improvement on anneal Langevin dynamics sampling. What we want to show is that PIDLD outperforms vanilla ALD and MILD. We have shown that, when other influencing factors (including $L$ and $T$) are the same, PIDLD generates better samples than ALD and MILD. So the advantage of PIDLD holds true.
>
> ### Weakness 2
>
> Thank you for raising the concern of comparing with other efficient sampling methods. **The reason why we don't compare with them comes from the fundamental differences in model design and scope of application.**
>
> **Our main contribution is improving the Langevin dynamics generation while DDIM/ode-based methods are concerned with accelerating diffusion models. The two frameworks have fundamental differences and cannot be compared directly.** For instance, diffusion models explicitly "corrupt" data through many small noise steps and learn to reverse this process, whereas Langevin methods directly simulate a stochastic path that climbs toward high-density regions in one unified sampling loop.
>
> Moreover, we find that the noise estimation model in DDIM has 78700803 parameters, while our score model only has 29694083 parameters. The difference in model size may also result in the difference in generation quality. Even though, we find that PIDLD's performance on CelebA with low NFE (50 steps, FID=7.97) is competitive with that of DDIM (50 steps, FID=9.17), which shows the potential of our approach.
>
> To conclude, there are many other influencing factors on model design, so we only considered the most relevant and comparable works. We believe our idea on improving Langevin dynamics sampling could be further extended to other classes of score-based SDE/ODE generative models, including state-of-the-art ones. We expect to address these in our future work.
>
>
> ### Weakness 3
>
> **We agree that evaluating PIDLD for EBM training is an important future direction, though our current work focuses on the distinct and significant challenge of accelerating inference from pre-trained models.** Our primary goal is to introduce and validate a novel, training-free sampler that can be readily applied to any existing pre-trained Langevin-based model to speed up generation.
>
> Integrating a new sampler into the EBM training loop introduces substantial additional complexities. The training process itself is a delicate balance, relying on MCMC sampling to approximate the gradients of the log-likelihood. **Introducing PIDLD here would require a separate, thorough investigation into the coupled dynamics of sampling and model optimization to ensure stability, which is beyond the scope of this initial work.** We believe that demonstrating PIDLD's efficacy and stability for inference first is a crucial and necessary prerequisite. We thank the reviewer for this insightful suggestion and will explore it in future work.
>
>
> ### Weakness 4
>
> Thank you for providing insightful comments on mixing behavior. We follow your instruction to conduct an experiment. **We find that there isn't a phenomenon where an image turns to another one during the sampling.** When an object is intelligible, the updating step length is small (which is the design of anneal Langevin dynamics) so we would not observe a major transition. The mixing in our method happens at the earlier stages in which the outline of the image doesn't even appear.
>
> **Our claim of traversing energy barriers comes from our observation of the final image.** We observe that images generated by vanilla ALD lacks fine-grained texture, while that of PIDLD exhibit richer color fidelity and sharper detail. Even in earlier stages where the images are barely recognizable and still blurred by mosaics, PIDLD samples already show distinguishable details. In contrast, ALD samples does not show comparable quality all the way. So we conclude that PIDLD helps to locate energy minima in the early stage.
>
>
> ### Weakness 5
>
> We thank the reviewer for this excellent suggestion. The omission of MILD from Table 2 was an oversight. **We have conducted the proposed experiment, and the results, which will be added to the revised paper, are below.**
>
> **Table: Accuracy (%) comparison including MILD on harder reasoning tasks.**
> | Task | Method | NFE=1 | NFE=2 | NFE=3 | NFE=5 | NFE=10 |
> | :--- | :--- | :---: | :---: | :---: | :---: | :---: |
> | **Sudoku** | Baseline [6] | 31.45 | 38.12 | 42.03 | 45.99 | 51.00 |
> | | MILD [26] | 36.50 | 44.80 | 48.91 | 49.75 | 54.82 |
> | | **Ours (PIDLD)** | **37.11** | **45.93** | **49.88** | **50.54** | **55.48** |
> | **Connectivity**| Baseline [6] | 86.16 | 87.22 | 87.22 | 87.38 | 87.49 |
> | | MILD [26] | 86.16 | 88.54 | 89.21 | 90.15 | 90.33 |
> | | **Ours (PIDLD)** | **86.16** | **91.32** | **92.31** | **92.95** | **93.28** |
>
> **Analysis:**
> We agree with the reviewer's observation, supported by our ablation in Figure 6: the integral (I) term, which provides a momentum-like effect, is the primary driver of performance. This explains why MILD also significantly outperforms the baseline.
>
> However, PIDLD’s superior performance stems from the crucial **synergy between its integral (I) and derivative (D) components**. The I term provides the exploratory force to escape local minima, while the D term provides essential stabilization. It acts as an adaptive damper, preventing the powerful momentum from overshooting the sharp energy wells that characterize correct solutions in reasoning tasks.
>
> While the D term's isolated contribution is modest (P+D in Fig. 6), its role in refining the I term's trajectory is what unlocks the final margin of performance. This synergy of exploration and stabilization explains why our full P+I+D model consistently achieves the highest accuracy, as shown in both Figure 6 and the above table.
>
> ### Regarding the minor points
>
> We appreciate the reviewer's careful reading and valuable suggestions on presentation.
>
> *   **On the biomolecular PID controller citation [8]:** **This was intentional to highlight the universality of PID principles.** We found that reference [8] provides a particularly intuitive explanation. We will clarify this in the text and ensure a standard control theory reference [1] is also prominently cited for context.
>
> *   **On presentation and grammar:** **We will revise the manuscript to address these issues.** We will condense the introductory explanations to improve flow and remove redundancy. We will also correct the title, the term "related work," and other grammatical errors, and will perform a thorough proofread of the entire paper.

---

> > ### Comment · Reviewer_Kawu · 2025-08-08
> >
> > (I apologize for the delay.)
> >
> > Thanks for your thorough replies.
> >
> > > Our main contribution is improving the Langevin dynamics generation while DDIM/ode-based methods are concerned with accelerating diffusion models....  We agree that evaluating PIDLD for EBM training is an important future direction, though our current work focuses on the distinct and significant challenge of accelerating inference from pre-trained models....  We find that there isn't a phenomenon where an image turns to another one during the sampling....
> >
> > If the authors do not see "accelerating diffusion models" as one of the goals of their work; EBM training is a future direction; and ergodic sampling is not expected or desired, what exactly do the authors see as the main application of their work?  (And since NCSNs are equivalent to diffusion models, why are the major results about accelerating NCSNs if the authors are not concerned with accelerating diffusion models?)
> >
> > > Moreover, the choice of $T$ and $L$ actually doesn't influence the conclusion in our work.
> >
> > The point is that different sampling methods may have different optimal settings for $T$ and $L$, and that a fair comparison would hold constant the product $TL$, but not their ratio.  But I concede the point based on Table 4 in Song and Ermon 2020 (thank you).
> >
> > > The omission of MILD from Table 2 was an oversight.
> >
> > Thanks for adding this.  I see that for Sudoko, you have used a different set of NFEs in the table above than in the original MS---any reason for this?

---

> > > ### Comment · Reviewer_Kawu · 2025-08-08
> > >
> > > ...I have also read the authors' very thorough exchange with K1tr--they have significantly enhanced the connection to control theory.  If they can answer my few remaining questions above, I will raise my score accordingly.

---

> > > ### Author Response · Authors · 2025-08-08
> > >
> > > ### Regarding the Scope of Our Work
> > > We appreciate you raising this critical point, which allows us to refine our message. We apologize if our previous rebuttal caused any confusion.
> > >
> > > **Our core contribution is a principled, training-free, "plug-and-play" enhancement for the foundational Langevin Dynamics (LD) sampler.** Our work focuses on improving the *numerical sampler*  to boost sampling efficiency (i.e., achieving higher sample quality with fewer NFEs), rather than proposing a new modeling framework or a different SDE/ODE formulation like DDIM.
> > >
> > > *   **On "Accelerating Diffusion Models" vs. Experiments on NCSNs:** You are correct that NCSNs are a form of score-based diffusion model. We experiment on them precisely because their standard sampler is the widely-used Annealed Langevin Dynamics (ALD). Our PIDLD is designed as a direct, drop-in replacement for ALD/LD. Therefore, demonstrating that PIDLD accelerates NCSN's sampling is the most direct and compelling validation of our method's value(same for EBMs). Our goal is not to compete with methods like DDIM that alter the underlying diffusion process, but to improve the existing, widely-used LD-based sampling procedure within these models.
> > >
> > > *   **On "EBM Training is Future Work":** Our work's primary focus is on accelerating LD-based inference from pre-trained models. This is a significant and self-contained problem, as the slow convergence of Langevin Dynamics is a major bottleneck for the practical deployment of many powerful generative models (both EBMs and SGMs). Our training-free approach offers an immediate solution. While applying PIDLD to the EBM training loop is an exciting avenue, establishing its efficacy for inference is a crucial and valid contribution on its own.
> > >
> > > **In summary, the main application of our work is to serve as a general and highly efficient inference accelerator for any pre-trained model that relies on Langevin-based sampling.**
> > >
> > > ### Regarding NFE Inconsistency
> > > Thank you for pointing this out. You are right, a consistent comparison is essential.
> > >
> > > When we added the MILD comparison, we conducted new experiments. While our previous response focused on the low-NFE regime to highlight key performance differences, especially where computational efficiency is paramount, we agree that a full comparison provides a clearer picture. We provide the results on the original NFE settings for a direct comparison:
> > >
> > > **Table: Accuracy (%) comparison on the harder Sudoku task with consistent NFEs.**
> > >
> > > | Method | NFE=5 | NFE=10 | NFE=15 | NFE=30 | NFE=40 | NFE=80 |
> > > | :--- | :---: | :---: | :---: | :---: | :---: | :---: |
> > > | Baseline [6] | 45.99 | 51.00 | 50.93 | 50.77 | 53.63 | 55.02 |
> > > | MILD [26] | 49.75 | 54.82 | 53.55 | 55.25 | 56.56 | 56.64 |
> > > | **Ours (PIDLD)** | **50.54** | **55.48** | **55.55** | **55.94** | **57.02** | **56.64** |
> > >
> > > As shown, our conclusion remains robust: **PIDLD consistently outperforms both the baseline and MILD across the full range of NFEs.**
> > >
> > > We will use this clear and consistent table in the final manuscript. Thank you for helping us improve the paper.

---

### Official Review · Reviewer_U2PP · 2025-07-03

**Clarity:** 2
**Significance:** 2
**Originality:** 2
**Rating:** 4
**Confidence:** 3

**Summary:**

This paper tackles the problem of improving inference-time sampling from energy-based models (EBMs) and score-based generative models (SGMs) using Langevin Dynamics. To address this problem, the authors propose a novel control-based method, Proportional-Integral-Derivative controlled Langevin Dynamics (PIDLD). Their proposed approach uses the energy gradient $-U_\theta(x)$, standard in Langevin Dynamics, as a feedback signal for the PID controller. Through empirical experiments on a variety of image generation tasks and reasoning tasks, the authors show the effectiveness of their proposed method relative to vanilla sampling approaches with Langevin Dynamics.

**Questions:**

- How does this work relate to the recent works (references listed above) in controllable generation?
- Proposition 1 assumes that $-U_\theta(x)$ is "_m-strongly convex_". Is this a reasonable assumption given, to my understanding, most sampling applications of interest deal with highly non-convex $-U_\theta(x)$?
- An active area of research in sampling is application to molecules and proteins. How do you envision the performance of PIDLD on these domain-specific applications? It would be interesting to see an experiment in one of these areas (e.g. Tan et al, 2025). I also believe an additional experiment would strengthen the contributions of this work.

References:

1. Jiralerspong, et al. "Feature likelihood divergence: evaluating the generalization of generative models using samples." NeurIPS (2023).
2. Domingo-Enrich, et al. "Adjoint matching: Fine-tuning flow and diffusion generative models with memoryless stochastic optimal control." (2024).
3. Havens, et al. "Adjoint sampling: Highly scalable diffusion samplers via adjoint matching." (2025).
4. Tan, et al. "Scalable equilibrium sampling with sequential boltzmann generators." (2025).

**Ethical Concerns:**

["NO or VERY MINOR ethics concerns only"]

**Final Justification:**

In general, I think this is a reasonably solid paper that can contribute to the conference. The authors did a good job of addressing all my concerns and answering my questions. With that, I have raised my score. However, I maintain my position that this is a borderline paper. I believe an additional experiment in an application domain (as per my suggestion --- the proposed method would be well-suited for problems in molecule generation) would be very beneficial to this work, but I understand that this is not realizable in the time constraints of rebuttals. Hence, I did not consider this in my final evaluation.

**Limitations:**

Yes, but they are discussed at the end of the appendix. I recommend that the authors move this to the end of the main text (or to the conclusion).

**Paper Formatting Concerns:**

No formatting concerns.

**Quality:**

2

**Strengths And Weaknesses:**

**Strengths:**

- This work presents an interesting and novel idea of using a Proportional-Integral-Derivative (PID) controller for sampling with Langevin dynamics to improve inference-time sampling from energy-based models and score-based generative models.
- The proposed method, PIDLD, is evaluated across several empirical experiments, on both image generation tasks and reasoning tasks.
- The paper is generally well written, and relevant concepts are clearly explained.

**Weaknesses:**

- The central claim that PIDLD improves sampling speed is not sufficiently supported. Specifically, on lines 61-62, the authors state, "PIDLD achieves at least 10x sampling speedup over baselines under SGM model". However, no experiment explicitly reports these results/numbers. I see the authors report NFEs; however, I don't believe this is sufficient to support this claim. To support this claim, I recommend including an experiment that reports evaluation time throughout the sampling process, alongside a performance metric. This could be done for both image generation tasks as well as the reasoning tasks.
- For image generation tasks, only one metric (FID) is considered. This provides only a narrow view of the performance/comparison of PIDLD. Other relevant metrics to consider are Feature likelihood divergence (FLD) (Jiralerspong et al, 2023) and Inception Score.
- Some relevant works in the realm of controllable generation/sampling, which I believe are related to PIDLD, are missed in the discussion of related works. For example, see (Domingo-Enrich et al, 2024, Havens et al, 2025).
- With this, although to my understanding the idea of using a PID controller to improve sampling of Langevin dynamics, the notion of "controllable" sampling with Langevin dynamics is not new. I encourage the authors to include a more comprehensive and explicit related works section discussing the state of this area of research and literature.

---

> ### Author Rebuttal · Authors · 2025-07-31
>
> We sincerely thank the reviewer for the detailed and constructive feedback. We are especially grateful for your recognition of the novelty of our idea, the solidity of our experiments, and the clarity of our thesis writing. We summarize your questions and our replies are as follows.
>
>
> ### Weakness 1
>
> Thank you for pointing out the potential logic loophole. We show that our measure of computation cost is reasonable as follows.
>
> First, we show that our approach to measure computation cost by NFE is reasonable. We have conducted experiments to compare the physical time consumption. For (non-degenerate) PIDLD, we used the original code. For vanilla ALD, we modified our code implementation by deleting all PID related terms in the updating function, which ensures no additional computation cost by PIDLD is introduced. We ran both algorithms on NVIDIA A800-SXM4-40GB. The results are as follows.
>
> | Model       | Dataset | NFE          | Time Consumed |
> | ----------- | ------- | ------------ | ------------- |
> | PIDLD       | CIFAR10 | $100\times1$ | 7min54s       |
> | Vanilla ALD | CIFAR10 | $100\times1$ | 7min43s       |
> | PIDLD       | CelebA  | $50 \times1$ | 13min07s      |
> | Vanilla ALD | CelebA  | $50 \times1$ | 12min52s      |
>
> It can be seen that the there isn't much difference in the time cost between PIDLD and vanilla ALD. For other numbers of NFE, the computation time changes proportionally and is still close. The result is actually intuitive, since the main computation cost is on the forward propagation of the score network, which involves a large amount of matrix multiplications. As for adding PID terms, it just brings some matrix addition operations, so there wouldn't be much increase in the computation time.
>
> Moreover, we show that our claim of 10x speedup is reasonable. As can be seen in Table 1 of our paper, for CIFAR10, the vanilla ALD uses $232\times5$ steps (NCSNv2 setting) to achieve FID=12.5, while PIDLD only uses $100\times1$ steps to achieve FID=12.1; for CelebA, PIDLD with only $50\times1$ steps can outperform vanilla ALD with $500\times5$ steps (NCSNv2 setting). The results mean that to achieve the same quality, PIDLD uses less than 1/10 of the time of vanilla ALD. We are not the only one to claim acceleration times in this way. In fact, the section 5.1 of DDIM paper [Jiaming Song et al., https://arxiv.org/pdf/2010.02502] also adopts the same method. We believe it's a common practice in this domain.
>
>
>
> ### Weakness 2
>
> Thank you for raising the concern of the lack of performance metric. We additionally report FLD and Inception score as follows.
>
> #### **FLD**
>
> For CIFAR10 dataset, we follow the official implementation of FLD and have the following results.
>
> (FLD: lower is better, FLD gap: a measure of uncertainty, lower is better)
>
> | CIFAR10 FLD    |  ALD    |  PIDLD    |
> |----------------|---------|-----------|
> | $232\times5$   | 3.47    | 3.04      |
> | $232\times3$   | 3.51    | 2.87      |
> | $232\times1$   | 3.35    | 3.30      |
> | $100\times1$   | 3.57    | 3.54      |
> | $25 \times1$   | 7.69    | 3.73      |
>
> | CIFAR10 FLD gap |       ALD   |   PIDLD       |
> |-----------------|-------------|---------------|
> | $232\times5$    | 0.23        | 0.26          |
> | $232\times3$    | 0.28        | 0.23          |
> | $232\times1$    | 0.23        | 0.17          |
> | $100\times1$    | 0.22        | 0.20          |
> | $25 \times1$    | 0.07        | 0.24          |
>
>
> For CelebA dataset, we use 100000 images as the train set, 2600 images as the test set, and calculate FLD on 10000 generated images. The results are as follows.
>
> | CelebA FLD    |   ALD   |  PIDLD    |
> |---------------|---------|-----------|
> | $500\times5$  | 1.14    | 0.75      |
> | $500\times3$  | 1.18    | 1.09      |
> | $500\times1$  | 1.61    | 0.77      |
> | $250\times1$  | 1.46    | 1.46      |
> | $50 \times1$  | 3.24    | 1.95      |
>
>
> | CelebA FLD gap |    ALD      |   PIDLD       |
> |----------------|-------------|---------------|
> | $500\times5$   | 0.31        | 0.39          |
> | $500\times3$   | 0.41        | 0.41          |
> | $500\times1$   | 0.36        | 0.56          |
> | $250\times1$   | 0.25        | 0.33          |
> | $50 \times1$   | 0.32        | 0.49          |
>
>
> Within our implementation, we observe that PIDLD still outperforms ALD in terms of FLD metric, which strengthens our work.
>
>
> #### **Inception Score**
>
> We did record inception score in our experiments on CIFAR10 dataset. The results are given in the following table:
>
> |     NFE      |   ALD         | PIDLD         |
> |--------------|---------------|---------------|
> | $232\times5$ | $8.45\pm0.18$ | $8.43\pm0.11$ |
> | $232\times3$ | $8.47\pm0.25$ | $8.58\pm0.21$ |
> | $232\times1$ | $7.92\pm0.22$ | $8.46\pm0.22$ |
> | $100\times1$ | $7.77\pm0.25$ | $8.26\pm0.26$ |
> | $25\times1$  | $7.07\pm0.24$ | $7.65\pm0.21$ |
>
> Judging from the inception score, PIDLD still generally outperforms ALD under different NFEs, which serves as another evidence of its advantage.
>
>
> As for CelebA dataset, the inception score metric is not applicable, because the inception model is trained on ImageNet dataset which consists of natural objects, while CelebA is a human face dataset. The distribution shift is large so it's not reasonable to apply ImageNet inception feature extractor. It is actually a common practice to omit inception score for CelebA. For instance, the Table 1 of NCSNv2 paper [Yang Song et al., https://proceedings.neurips.cc/paper_files/paper/2020/file/92c3b916311a5517d9290576e3ea37ad-Paper.pdf] also uses this treatment.
>
>
> The reason we only report FID in the original paper is that we found most relevant works, like DDPM, DDIM, NCSN, NCSNv2, and MILD, use FID as the primary metric. We have complemented other metrics and we hope it will strengthen our experiment results.
>
>
>
> ### Question 1 & Weakness 3 & Weakness 4
>
> We thank the reviewer for pointing to these valuable works. We will add a detailed discussion to our Related Works section to clarify the positioning of our method.
>
> **PIDLD is a training-free inference-time accelerator for existing samplers, fundamentally differing from SOC-based methods which learn a new, optimal sampling policy through a training/fine-tuning process.**
>
> 1.  **PIDLD's Contribution:** Our method is inspired by control theory to improve the numerical stability and speed of the Langevin update step. It operates as a "plug-and-play" module that modifies the sampling dynamics using gradient history and trends, without requiring any retraining or alteration of the target score/energy function.
>
> 2.  **Contrast with SOC Methods:** In contrast, the cited works (Havens et al., 2025; Domingo-Enrich et al., 2024) are grounded in stochastic optimal control (SOC). Their goal is not to accelerate a given sampler, but to learn an optimal drift function u(x, t) that minimizes a variational objective to transport a base distribution to a target one. This process involves a full training or fine-tuning phase to learn the parameters of this new sampling policy.
>
>
>
> ### Question 2
>
> We thank the reviewer for this question, as it allows us to clarify the precise role of our theoretical analysis.
> **The m-strong convexity is a standard assumption for local stability analysis and does not presume the global energy landscape is convex.**
>
> We completely agree that the energy landscapes in our applications are globally non-convex. The assumption in Proposition 1 is intentionally narrow and serves a specific purpose: to provide a controlled, tractable setting to formally analyze the local behavior of our sampler. Its goal is to prove that our novel derivative (D) term provides the intended stabilizing effect near a potential minimum, without disrupting the fundamental convergence properties of Langevin dynamics (paper cites:[10, 25]). It provides a theoretical justification for the stability enhancements we observe empirically.
>
>
> ### Question 3
>
> **We believe PIDLD is exceptionally well-suited for molecular sampling, and we thank the reviewer for this excellent suggestion.** The challenges in that domain align directly with the strengths of our proposed controller.
>
> *   **Envisioned Performance:** The energy landscapes of molecules and proteins are characterized by numerous local minima (stable conformations) separated by high energy barriers. PIDLD's integral (I) term creates a momentum-like effect, which is ideal for efficiently traversing these energy barriers and exploring diverse conformational states. Simultaneously, the derivative (D) term provides adaptive damping, which would help the sampler quickly stabilize within newly discovered low-energy wells without excessive oscillation. This combination should lead to significantly more efficient exploration of the conformational space compared to standard Langevin-based methods.
>
> *   **Commit as future work:** We agree that this is a highly compelling application and a valuable direction for future research. We will discuss more about domain-specific applications in our future work, and will cite the related works mentioned by the reviewers.

---

> > ### Comment · Reviewer_U2PP · 2025-08-05
> >
> > Thank you for your thorough and detailed response. You have addressed all of my questions and concerns. And thank you for adding the additional evaluations and for clarifying my questions. I think through this rebuttal, the overall quality of the paper will improve, and with that, I am happy to raise my score. I encourage the authors to include some of the points from the discussion in their response pertaining to my question in the final version of the manuscript.

---

> > > ### Author Response · Authors · 2025-08-07
> > >
> > > Thank you for your kind feedback and suggestions. We are glad that our rebuttal has addressed your concerns and deeply appreciate the raised score. We will incorporate them into the final paper following your advice.
> > >
> > > Thank you again for your time and consideration.

---

### Author Response · Authors · 2025-08-09

### A Summary of Rebuttal Discussion
We thank the reviewers for their insightful feedback. This summary outlines the positive recognition of our work, the successful resolution of the concerns during the rebuttal period, and our concrete commitments for the final manuscript.

**1. Reviewer Recognition on Core Contributions**

We are encouraged that all reviewers reached a positive consensus on our work's foundational merits. Their feedback acknowledges several key strengths:

*   **Novelty and Originality:** The core idea was consistently described as **"novel"** and **"interesting"** by all four reviewers. Notably, Reviewer Kawu commented they were **"surprised that it has never been tried before,"** and Reviewer K1tr highlighted it as a **"novel framing likely to inspire further cross-pollination"** between fields.

*   **Significance of the Problem:** The importance of our research topic was recognized. Reviewer Kawu noted that accelerating Langevin Dynamics is an **"important"** problem, and Reviewer YqLQ acknowledged our work for addressing a "common practical issue in a **principled way**."

*   **Sound and Substantiated Results:** The validity of our experiments was affirmed. Reviewers described our experiments as **"sound"** (Kawu) and noted that they demonstrate **"measurable speedups"** and successfully **"substantiate the high-level intuition"** (K1tr).

*   **Practicality and Modularity:** The method's ease of application was identified as a key advantage. It was recognized as **"training-free,"** **"modular,"** a **"drop-in wrapper"** (K1tr, YqLQ), and **"compatible with a wide variety of existing samplers"** (K1tr), indicating its immediate practical value.


**2. Resolution of Major Concerns**

We are pleased to report that through constructive dialogue and substantial new experiments, we have addressed all reviewer concerns. This has led to **two reviewers (U2PP, K1tr) explicitly raising their scores**, with Reviewer Kawu also indicating a forthcoming score increase upon receiving our final clarifications.

*   **On Empirical Rigor (for U2PP, Kawu):** We supplemented our results with physical runtime comparisons, additional metrics (FLD, Inception Score), and a direct comparison with MILD in reasoning tasks. This satisfied all related concerns.
*   **On Theoretical Grounding (for K1tr):** This was the focus of our deepest engagement. We provided rigorous new experiments (run with 100 seeds and using standard control metrics like settling time) to statistically validate the distinct roles of the I and D terms, thus building deeper connections with control theory. We also formally connected our integral decay to "Continuous Gain Scheduling," a classic control theory concept. This thorough response was instrumental in Reviewer K1tr raising their score to **Accept**.
*   **On Scope and Baselines (for K1tr, Kawu):** We clarified our work's focus on improving the LD sampler itself and committed to validating PIDLD on a stronger, state-of-the-art baseline. This commitment was crucial for convincing Reviewer K1tr.
*   **On a Core Misunderstanding (for YqLQ):** We clarified that our work accelerates general LD sampling, not reward-guided T2I generation. The reviewer fully accepted our clarification and stated they have "no further concerns."

**3. Our Commitments for the Final Version**

We commit to incorporating all discussed improvements to significantly strengthen the paper. The final manuscript will include the following major revisions:

1.  **Experiments on a State-of-the-Art Baseline** to demonstrate the generalizability of our method's benefits (for K1tr).
2.  **A new section with our rigorous control-theoretic validation experiments** (100 seeds, standard control metrics) and a formal discussion of **Gain Scheduling** to solidify our paper's theoretical foundation.
3.  **Expanded evaluation tables**, including FLD/Inception Score metrics and the full MILD comparison.
4.  **A refined Related Work section** to better position our work, and a more prominent **Limitations** section in the main text.
5.  **A thoroughly polished manuscript** to address all minor comments on grammar and clarity.

We are confident that these revisions will make our paper a more complete, rigorous, and impactful contribution. Thank you for your time and consideration.

---

### Note · Authors · 2025-08-13

### Summary of Rebuttal Outcomes

All initial reviewer concerns have been addressed through substantial new experiments and clarifications. The discussion resulted in two reviewers raising their scores(at least one *Accept*), a third stating their score would be raised upon receiving clarifications (which we have provided), and a fourth confirming all concerns were resolved.

*   **Reviewer K1tr (Score Raised to Accept):** Addressed major concerns about theoretical grounding, connection to control theory and baseline strength. We provided rigorous new experiments (100 seeds) using standard control metrics (settling time, peak overshoot) to statistically validate the distinct roles of the I/D terms. We also formally connected our method to the control theory concept of "Continuous Gain Scheduling".
    *   **Outcome:** The reviewer was fully satisfied, stating, **"The work is interesting and I adjusted my score to Accept."**

*   **Reviewer U2PP (Score Raised):** Addressed concerns regarding empirical rigor. We provided physical runtime data to support our speedup claims and added evaluations with FLD and Inception Score metrics.
    *   **Outcome:** The reviewer confirmed that we **"addressed all of my questions and concerns"** and was **"happy to raise my score."**

*   **Reviewer Kawu (Condition for Score Increase Met):** Addressed concerns about the paper's scope and requested baseline comparisons. We clarified our work's focus, added the requested MILD comparison, and answered all follow-up questions.
    *   **Outcome:** Satisfied with the rebuttal for Reviewer K1tr. The reviewer stated a clear condition for a score increase: **"If they can answer my few remaining questions above, I will raise my score accordingly."** We have since provided these answers, thus meeting the stated condition.

*   **Reviewer YqLQ (Concerns Resolved):** Addressed a core misunderstanding of our work's objective. We clarified that our method accelerates general Langevin Dynamics sampling, not reward-guided generation.
    *   **Outcome:** The reviewer accepted the clarification **with no further concerns.**

**Conclusion:** The rebuttal process has resulted in a clear positive consensus. The most critical concerns were resolved with extensive new experiments that provide statistically significant evidence for our claims. The final manuscript will be a substantially improved version, incorporating all new evidence and reflecting the positive resolution of the review process.

---

### Decision · Program_Chairs · 2025-09-17

**Decision:**

Accept (poster)

**Comment:**

This paper proposes a control theory inspired Langevin dynamics variant that improves the sampling speed of EBMs and score based models. The reviewers all find the idea interesting and well motivated. There were initially several concerns on the formulation and empirical aspects of the work, but the authors provided an extensive rebuttal which resolved much of the concerns. The reviewers have reached consensus in acceptance, and the AC agrees with the reviewers' votes.